# Notch and TLR signaling coordinate monocyte cell fate and inflammation

Jaba Gamrekelashvili[1,2]*, Tamar Kapanadze[1,2], Stefan Sablotny[1,2], Corina Ratiu[3], Khaled Dastagir[1,4], Matthias Lochner[5,6], Susanne Karbach[7,8,9], Philip Wenzel[7,8,9], Andre Sitnow[1,2], Susanne Fleig[1,2], Tim Sparwasser[10], Ulrich Kalinke[11,12], Bernhard Holzmann[13], Hermann Haller[1], Florian P Limbourg[1,2]*

[1]Vascular Medicine Research, Hannover Medical School, Hannover, Germany; [2]Department of Nephrology and Hypertension, Hannover Medical School, Hannover, Germany; [3]Institut für Kardiovaskuläre Physiologie, Fachbereich Medizin der Goethe-Universität Frankfurt am Main, Frankfurt am Main, Germany; [4]Department of Plastic, Aesthetic, Hand and Reconstructive Surgery, Hannover Medical School, Hannover, Germany; [5]Institute of Medical Microbiology and Hospital Epidemiology, Hannover Medical School, Hannover, Germany; [6]Mucosal Infection Immunology, TWINCORE, Centre for Experimental and Clinical Infection Research, Hannover, Germany; [7]Center for Cardiology, Cardiology I, University Medical Center of the Johannes Gutenberg-University Mainz, Mainz, Germany; [8]Center for Thrombosis and Hemostasis, University Medical Center of the Johannes Gutenberg-University Mainz, Mainz, Germany; [9]German Center for Cardiovascular Research (DZHK), Partner Site Rhine Main, Mainz, Germany; [10]Department of Medical Microbiology and Hygiene, Medical Center of the Johannes Gutenberg-University of Mainz, Mainz, Germany; [11]Institute for Experimental Infection Research, TWINCORE, Centre for Experimental and Clinical Infection Research, a joint venture between the Helmholtz Centre for Infection Research Braunschweig and the Hannover Medical School, Hannover, Germany; [12]Cluster of Excellence-Resolving Infection Susceptibility (RESIST), Hanover Medical School, Hannover, Germany; [13]Department of Surgery, Klinikum rechts der Isar, Technical University Munich, Munich, Germany

*For correspondence:
Gamrekelashvili.Jaba@mh-hannover.de (JG);
limbourg.florian@mh-hannover.de (FPL)

**Competing interests:** The authors declare that no competing interests exist.

**Abstract** Conventional Ly6C$^{hi}$ monocytes have developmental plasticity for a spectrum of differentiated phagocytes. Here we show, using conditional deletion strategies in a mouse model of Toll-like receptor (TLR) 7-induced inflammation, that the spectrum of developmental cell fates of Ly6C$^{hi}$ monocytes, and the resultant inflammation, is coordinately regulated by TLR and Notch signaling. Cell-intrinsic Notch2 and TLR7-Myd88 pathways independently and synergistically promote Ly6C$^{lo}$ patrolling monocyte development from Ly6C$^{hi}$ monocytes under inflammatory conditions, while impairment in either signaling axis impairs Ly6C$^{lo}$ monocyte development. At the same time, TLR7 stimulation in the absence of functional Notch2 signaling promotes resident tissue macrophage gene expression signatures in monocytes in the blood and ectopic differentiation of Ly6C$^{hi}$ monocytes into macrophages and dendritic cells, which infiltrate the spleen and major blood vessels and are accompanied by aberrant systemic inflammation. Thus, Notch2 is a master regulator of Ly6C$^{hi}$ monocyte cell fate and inflammation in response to TLR signaling.

## Introduction

Infectious agents or tissue injury trigger an inflammatory response that aims to eliminate the inciting stressor and restore internal homeostasis (*Bonnardel and Guilliams, 2018*). The mononuclear phagocyte system (MPS) is an integral part of the inflammatory response and consists of the lineage of monocytes and macrophages (MF) and related tissue-resident cells. A key constituent of this system are monocytes of the major (classic) monocyte subtype, in mice called Ly6C$^{hi}$ monocytes. They originate from progenitor cells in the bone marrow (BM), circulate in peripheral blood (PB) and respond dynamically to changing conditions by differentiation into a spectrum of at least three distinct MPS effector phagocytes: Macrophages, dendritic cells (DC), and monocytes with patrolling behavior (*Arazi et al., 2019*; *Bonnardel and Guilliams, 2018*; *Chakarov et al., 2019*; *Gamrekelashvili et al., 2016*; *Hettinger et al., 2013*). The diversity of monocyte differentiation responses is thought to be influenced by environmental signals as well as tissue-derived and cell-autonomous signaling mechanisms to ensure context-specific response patterns of the MPS (*Okabe and Medzhitov, 2016*). However, the precise mechanisms underlying monocyte cell fate decisions under inflammatory conditions are still not fully understood.

When recruited to inflamed or injured tissues, Ly6C$^{hi}$ monocytes differentiate into MF or DC with a variety of phenotypes and function in a context-dependent-manner and regulate the inflammatory response (*Krishnasamy et al., 2017*; *Xue et al., 2014*). However, Ly6C$^{hi}$ monocytes can also convert to a second, minor subpopulation of monocytes with blood vessel patrolling behavior. In mice, these are called Ly6C$^{lo}$ monocytes and express CD43, CD11c and the transcription factors *Nr4a1*, *Pou2f2* (*Gamrekelashvili et al., 2016*; *Patel et al., 2017*; *Varol et al., 2007*; *Yona et al., 2013*). These monocytes have a long lifespan and remain mostly within blood vessels, where they crawl along the luminal side of blood vessels to monitor endothelial integrity and to orchestrate endothelial repair (*Auffray et al., 2007*; *Carlin et al., 2013*; *Getzin et al., 2018*). Steady-state monocyte conversion occurs in specialized endothelial niches and is regulated by monocyte Notch2 signaling activated by endothelial Notch ligands (*Avraham-Davidi et al., 2013*; *Bianchini et al., 2019*; *Gamrekelashvili et al., 2016*; *Varol et al., 2007*). Notch signaling is a cell-contact-dependent signaling pathway regulating cell fate decisions in the innate immune system (*Radtke et al., 2013*). Notch signaling regulates formation of intestinal CD11c$^+$CX$_3$CR1$^+$ immune cells (*Ishifune et al., 2014*), Kupffer cells (*Bonnardel et al., 2019*; *Sakai et al., 2019*) and macrophage differentiation from Ly6C$^{hi}$ monocytes in ischemia (*Krishnasamy et al., 2017*), but also development of conventional DCs (*Caton et al., 2007*; *Epelman et al., 2014*; *Lewis et al., 2011*), which is mediated by Notch2.

Toll-like receptor 7 (TLR7) is a member of the family of pathogen sensors expressed on myeloid cells. Originally identified as recognizing imidazoquinoline derivatives such as Imiquimod (R837) and Resiquimod (R848), TLR7 senses ssRNA, and immune-complexes containing nucleic acids, in a Myd88-dependent manner during virus defense, but is also implicated in tissue-damage recognition and autoimmune disorders (*Kawai and Akira, 2010*). TLR7-stimulation induces cytokine-production in both mouse and human patrolling monocytes and mediates sensing and disposal of damaged endothelial cells by Ly6C$^{lo}$ monocytes (*Carlin et al., 2013*; *Cros et al., 2010*), while chronic TLR7-stimulation drives differentiation of Ly6C$^{hi}$ monocytes into specialized macrophages and anemia development (*Akilesh et al., 2019*). Furthermore, systemic stimulation with TLR7 agonists induces progressive phenotypic changes in Ly6C$^{hi}$ monocytes consistent with conversion to Ly6C$^{lo}$ monocytes, suggesting involvement of TLR7 in monocyte conversion (*Santiago-Raber et al., 2011*). Here, we show that Notch signaling alters TLR-driven inflammation and modulates Ly6C$^{lo}$ monocyte vs. macrophage cell fate decisions in inflammation.

## Results

### TLR and Notch signaling promote monocyte conversion

We first studied the effects of TLR and/or Notch stimulation on monocyte conversion in a defined in vitro system (*Gamrekelashvili et al., 2016*). Ly6C$^{hi}$ monocytes isolated from the bone marrow (*Figure 1—figure supplement 1A and B*) of *Cx3cr1$^{gfp/+}$* reporter mice (GFP$^+$) were cultured with recombinant Notch ligand Delta-like 1 (DLL1) in the presence or absence of the TLR7/8 agonist R848 and analyzed after 24 hr for the acquisition of key features of Ly6C$^{lo}$ monocytes (*Gamrekelashvili et al.,*

*2016*; *Hettinger et al., 2013*). In contrast to control conditions, cells cultured with DLL1 showed an upregulation of CD11c and CD43, remained mostly MHC-II negative, and expressed transcription factors *Nr4a1* and *Pou2f2,* markers for Ly6C$^{lo}$ monocytes, leading to a significant, five-fold increase of Ly6C$^{lo}$ cells, consistent with enhanced monocyte conversion. Cells cultured with R848 alone showed a comparable phenotype response, both qualitatively and quantitatively (*Figure 1A–C*). Interestingly, on a molecular level, R848 stimulation primarily acted on *Pou2f2* induction and CD43 expression, while Notch stimulation primarily induced *Nr4a1* and CD11c upregulation. Furthermore, the combination of DLL1 and R848 strongly and significantly increased the number of CD11c$^+$CD43$^+$ Ly6C$^{lo}$ cells above the level of individual stimulation and significantly enhanced expression levels of both transcriptional regulators *Nr4a1* and *Pou2f2* (*Figure 1A–C*), suggesting in part synergistic and/ or cumulative regulation of monocyte conversion by TLR7/8 and Notch signaling. By comparison, the TLR4 ligand LPS also increased Ly6C$^{lo}$ cell numbers and expression levels of *Nr4a1* and *Pou2f2.* However, the absolute conversion rate was lower under LPS and there was no synergy/cumulative effect seen with DLL1 (*Figure 1D and E*).

Since monocyte conversion is regulated by Notch2 in vitro and in vivo (*Gamrekelashvili et al., 2016*), we next tested TLR-induced conversion in Ly6C$^{hi}$ monocytes with *Lyz2$^{Cre}$*-mediated conditional deletion of *Notch2* (*N2$^{\Delta My}$*). Both, littermate control (wt) and *N2$^{\Delta My}$* monocytes showed comparable response to R848, but conversion in the presence of DLL1, and importantly, also DLL1-R848 co-stimulation was significantly impaired in knock-out cells (*Figure 1F*). This suggests independent contributions of TLR and Notch signaling to monocyte conversion.

To study whether the TLR stimulation requires Myd88 we next tested purified Ly6C$^{hi}$ monocytes (*Figure 1—figure supplement 1C and D*) with Myd88 loss-of-function (*Myd88$^{-/-}$*). Compared to wt cells, *Myd88$^{-/-}$* monocytes showed strongly impaired conversion in response to R848 but a conserved response to DLL1. The response to DLL1-R848 co-stimulation, however, was significantly impaired (*Figure 1G*). Furthermore, expression of *Nr4a1* and *Pou2f2* by R848 was strongly reduced in *Myd88$^{-/-}$* monocytes with or without DLL1 co-stimulation, while DLL1-dependent induction was preserved (*Figure 1H*). Thus, Notch and TLR signaling act independently and synergistically to promote monocyte conversion.

To address the role of TLR stimulation for monocyte conversion in vivo we adoptively transferred sorted Ly6C$^{hi}$ monocytes from CD45.2$^+$GFP$^+$ mice into CD45.1$^+$ congenic recipients, injected a single dose of R848 and analyzed transferred CD45.2$^+$GFP$^+$ cells in BM and Spl after 2 days (*Figure 2A* and *Figure 1—figure supplement 1A and B*). Stimulation with R848 significantly promoted conversion into Ly6C$^{lo}$ monocytes displaying the proto-typical Ly6C$^{lo}$CD43$^+$CD11c$^+$MHC-II$^{lo/-}$ phenotype (*Figure 2B and C* and *Figure 2—figure supplement 1A*). In contrast, transfer of *Myd88$^{-/-}$* Ly6 C$^{hi}$ monocytes resulted in impaired conversion in response to R848 challenge (*Figure 2D and E* and *Figure 1—figure supplement 1D* and *Figure 2—figure supplement 1B*). Together, these data indicate that TLR and Notch cooperate in the regulation of monocyte conversion.

## Notch2-deficient mice show altered myeloid inflammatory response

To characterize the response to TLR stimulation in vivo, we applied the synthetic TLR7 agonist Imiquimod (IMQ, R837) in a commercially available crème formulation (Aldara) daily to the skin of mice (*El Malki et al., 2013*; *van der Fits et al., 2009*) and analyzed the systemic inflammatory response in control or *N2$^{\Delta My}$* mice (*Gamrekelashvili et al., 2016*; *Figure 3A*). While treatment with IMQ-induced comparable transient weight loss and ear swelling in both genotypes (*Figure 3—figure supplement 1A*), splenomegaly in response to treatment was significantly more pronounced in *N2$^{\Delta My}$* mice (*Figure 3—figure supplement 1B*).

To characterize the spectrum of myeloid cells in more detail, we next performed flow cytometry of PB cells with a dedicated myeloid panel (*Gamrekelashvili et al., 2016*) in wt or *N2$^{\Delta My}$ Cx3cr1$^{gfp/+}$* mice and subjected live Lin$^-$CD11b$^+$GFP$^+$ subsets to unsupervised t-SNE analysis (*Figure 3B*). This analysis strategy defined five different populations, based on single surface markers: Ly6C$^+$, CD43$^+$, MHC-II$^+$, F4/80$^{hi}$ and CD11c$^{hi}$ (*Figure 3C*). Applying these five gates to samples from separate experimental conditions identified dynamic alterations in blood myeloid subsets in response to IMQ, but also alterations in *N2$^{\Delta My}$* mice (*Figure 3D*). Specifically, abundance and distribution of Ly6C$^+$ cells, containing classical monocytes, in response to IMQ were changed to the same extend in both genotypes. In contrast, the MHC-II$^+$ and F4/80$^{hi}$ subsets were more abundant in *N2$^{\Delta My}$* mice, but also showed more robust changes in response to IMQ. On the other hand, the CD43$^+$ subset,

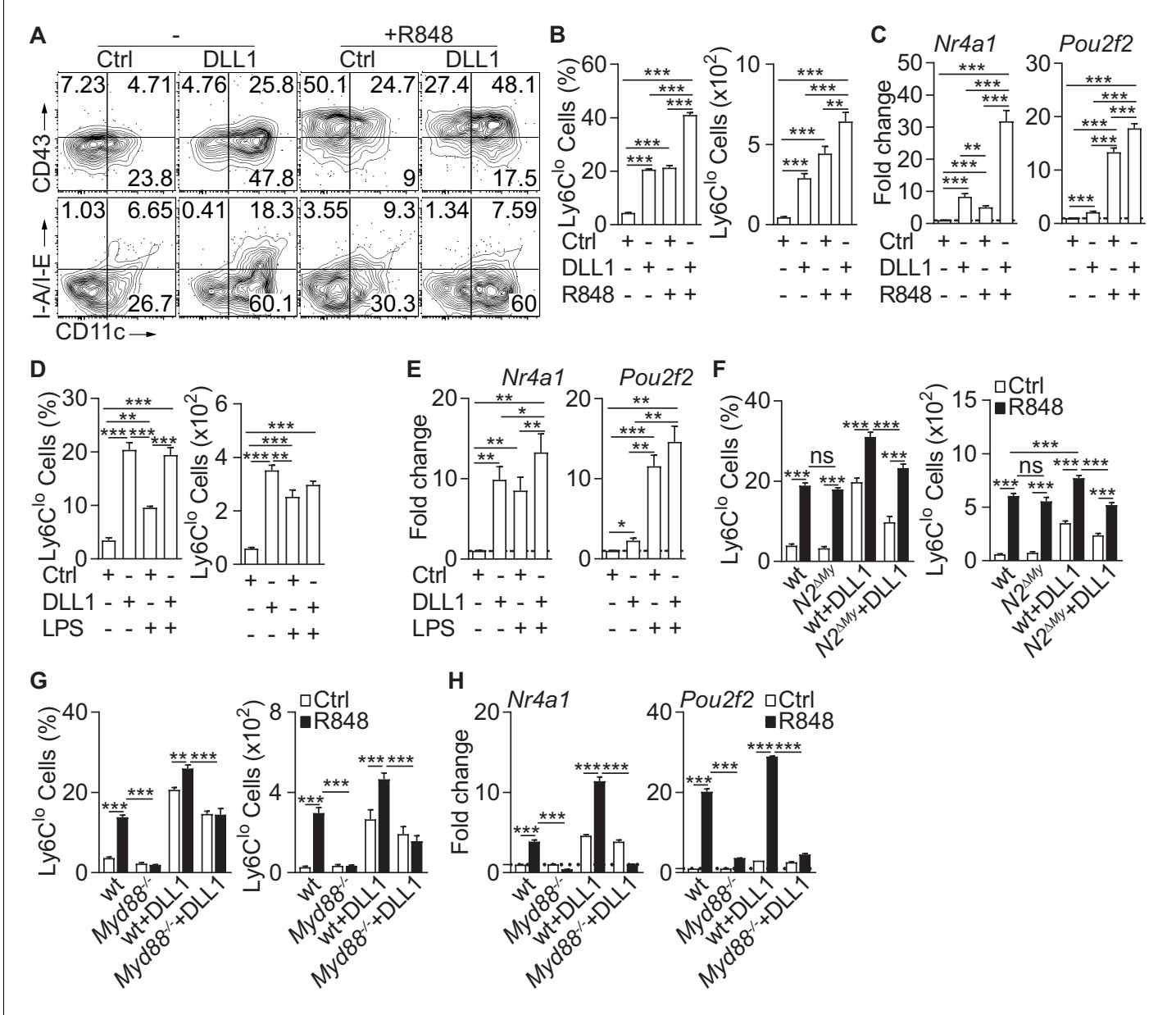

**Figure 1.** Inflammatory conditions enhance monocyte conversion in vitro. (A–F) Monocyte conversion in the presence of DLL1 and TLR agonists in vitro: (A) Representative flow cytometry plot, (B) relative frequency of Ly6C$^{lo}$ monocyte-like cells in live CD11b$^+$GFP$^+$ cells (left) or absolute numbers of Ly6C$^{lo}$ monocyte-like cells recovered from each well (right) are shown (representative of 3 experiments, n = 3). (C) Bar graphs showing expression of Ly6C$^{lo}$ monocyte hallmark genes, *Nr4a1* and *Pou2f2* from in vitro cultures treated with R848 (pooled from four experiments, n = 8–12). (D) Relative frequency (in live CD11b$^+$GFP$^+$ monocytes) or absolute numbers of Ly6C$^{lo}$ monocyte-like cells (from three experiments, n = 3) and (E) expression of *Nr4a1* and *Pou2f2* (from four experiments, n = 4–6) in the presence of LPS in vitro are shown. (F) wt or *N2$^{ΔMy}$* Ly6C$^{hi}$ monocyte conversion in the presence of DLL1 and R848 in vitro: relative frequency (in live CD11b$^+$GFP$^+$ monocytes) or absolute numbers of Ly6C$^{lo}$ monocyte-like cells (from three experiments, n = 4) is shown. (B, D, F) Absolute frequency of monocytes for Ctrl and DLL1 (in B, (D), and wt (Ctrl), wt+DLL1 (Ctrl) in (F) conditions are derived from the same experiments but are depicted as a three separate graphs for simplicity.(G, H) R848-enhanced conversion is *Myd88* dependent in vitro. Relative frequency (in live CD11b$^+$CX$_3$CR1$^+$ monocytes) or absolute numbers of Ly6C$^{lo}$ monocyte-like cells (G) and gene expression analysis in vitro (H) are shown (data are from two independent experiments, n = 3). (B, D, F–H) *p<0.05, **p<0.01, ***p<0.001; two-way ANOVA with Bonferroni's multiple comparison test. (C, E) *p<0.05, **p<0.01, ***p<0.001; paired one-way ANOVA with Geisser-Greenhouse's correction and Bonferroni's multiple comparison test.

The online version of this article includes the following figure supplement(s) for figure 1:

**Figure supplement 1.** Strategy of monocyte isolation from mouse bone marrow.

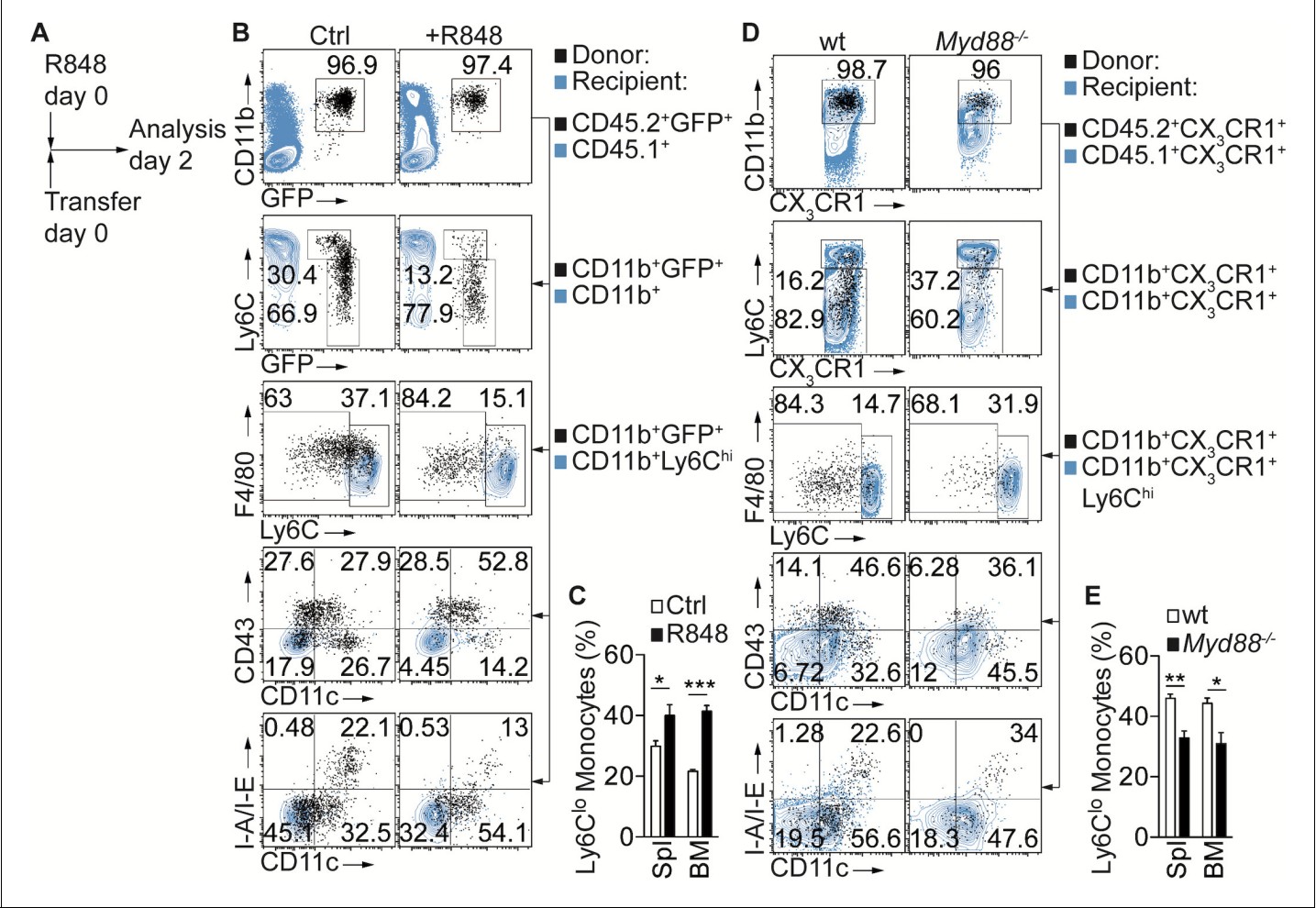

**Figure 2.** Inflammatory conditions enhance monocyte conversion in vivo. (A–E) Adoptive transfer and flow cytometry analysis of BM CD45.2$^+$ Ly6C$^{hi}$ monocytes in control or R848 injected CD45.1$^+$ congenic recipients: (A) Experimental setup is depicted; (B) Flow cytometry plots showing the progeny of transferred CD45.2$^+$CD11b$^+$Ly6C$^{hi}$CX$_3$CR1-GFP$^+$ (GFP$^+$) cells in black and recipient CD45.1$^+$ (1$^{st}$ row), CD45.1$^+$CD11b$^+$ (2$^{nd}$ row) or CD45.1$^+$CD11b$^+$Ly6C$^{hi}$ cells (3$^{rd}$ −5$^{th}$ rows) in blue; (C) Frequency of donor-derived Ly6C$^{lo}$ monocytes pooled from two independent experiments (n = 5). (D, E) R848-enhanced conversion is *Myd88* dependent in vivo. (D) Flow cytometry plots showing transferred CD45.2$^+$CX$_3$CR1$^+$ wt or *Myd88$^{-/-}$* cells in black and recipient CD45.1$^+$CX$_3$CR1$^+$ (1$^{st}$ row), CD45.1$^+$CX$_3$CR1$^+$CD11b$^+$ (2$^{nd}$ row) or CD45.1$^+$CX$_3$CR1$^+$CD11b$^+$Ly6C$^{hi}$ cells (3$^{rd}$ −5$^{th}$ rows) in blue. All recipient mice which received wt or *Myd88$^{-/-}$* donor cells were treated with R848; (E) Frequency of donor-derived Ly6C$^{lo}$ monocytes pooled from two independent experiments are shown (n = 4/5). (C, E) *p<0.05, **p<0.01, ***p<0.001; Student's *t*-test.
The online version of this article includes the following figure supplement(s) for figure 2:

**Figure supplement 1.** Inflammatory conditions enhance monocyte conversion in vivo.

containing the patrolling monocyte subset, showed prominent enrichment in wt mice, but was less abundant and showed diminished distribution changes after IMQ treatment in *N2$^{\Delta My}$* mice (*Figure 3D*).

To analyze the initially defined subsets more precisely, we applied a multi-parameter gating strategy to define conventional cell subsets (*Figure 3—figure supplement 2A and B* and *Supplementary file 1*; *Gamrekelashvili et al., 2016*).

In response to IMQ, Ly6C$^{hi}$ monocytes in wt mice increased transiently in blood, and this response was not altered in mice with conditional *Notch2* loss-of function (*Figure 3E*). In contrast, while Ly6C$^{lo}$ monocytes robustly increased over time with IMQ treatment in wt mice, their levels in *N2$^{\Delta My}$* mice were lower at baseline (*Gamrekelashvili et al., 2016*) and remained significantly reduced throughout the whole observation period (*Figure 3E and F* and *Figure 3—figure supplement 2A and B*). At the same time, while untreated *N2$^{\Delta My}$* mice showed increased levels of MHC-II$^+$

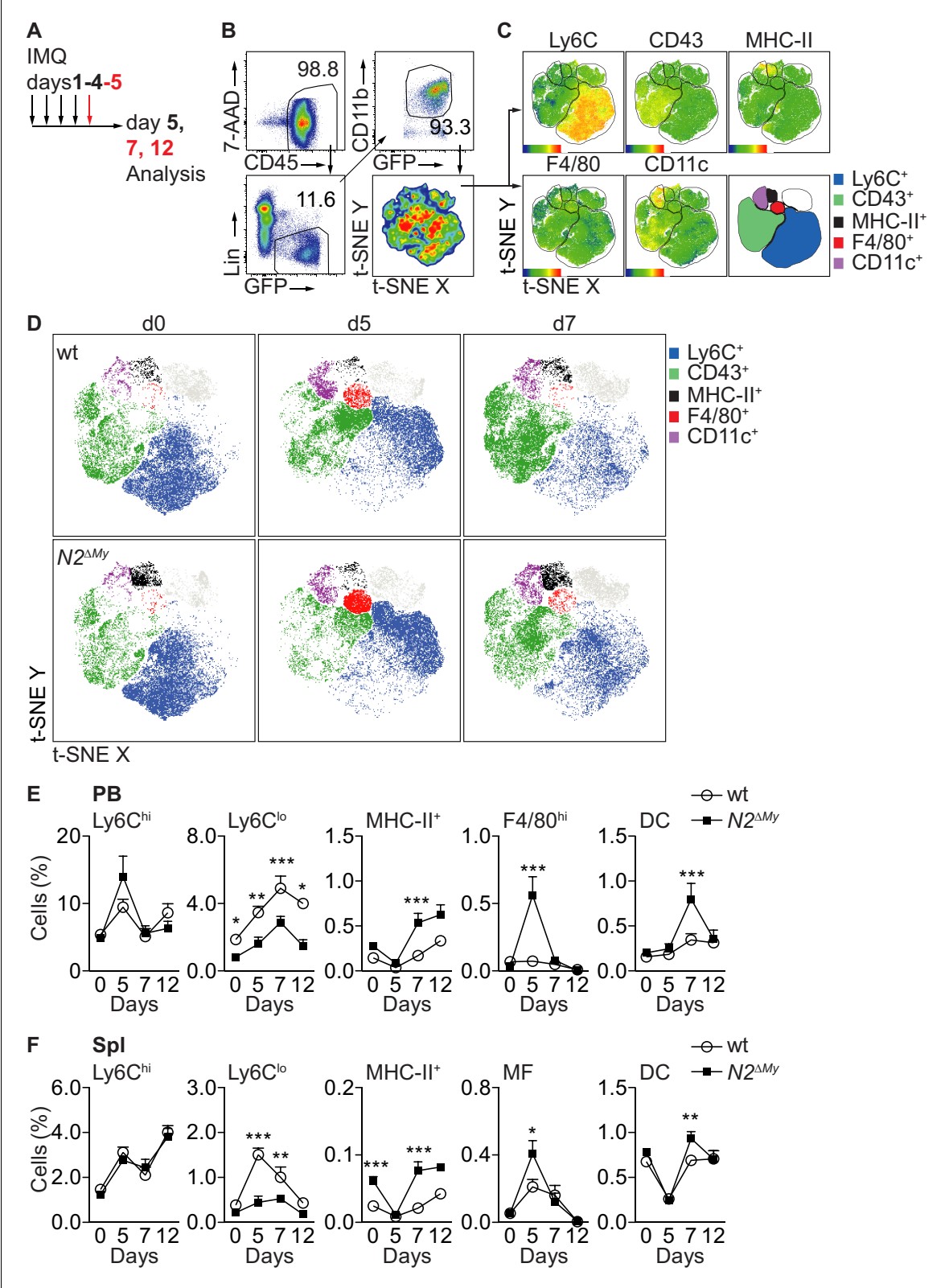

**Figure 3.** Acute inflammation triggers altered myeloid cell response in $N2^{\Delta My}$ mice. (A) Experimental set-up for IMQ treatment and analysis of mice. (B, C) Gating strategy for t-SNE analysis and definition of cell subsets based on expression of surface markers are shown. t-SNE was performed on live CD45+Lin-CD11b+GFP+ cells concatenated from 48 PB samples from four independent experiments. (D) Unsupervised t-SNE analysis showing composition and distribution of cellular subsets from PB of wt or $N2^{\Delta My}$ IMQ-treated or untreated mice at different time points defined in B, C) (n = 8

*Figure 3 continued on next page*

Figure 3 continued

mice are pooled for each condition). (E, F) Relative frequency of different myeloid subpopulations in PB and Spl of untreated or IMQ-treated mice are shown (data are pooled from six experiments n = 7–18). (E, F) *p<0.05, **p<0.01, ***p<0.001; two-way ANOVA with Bonferroni's multiple comparison test.

The online version of this article includes the following figure supplement(s) for figure 3:

**Figure supplement 1.** IMQ treatment induces systemic inflammation in mice.
**Figure supplement 2.** Identification of myeloid cell subsets in IMQ-treated mice.
**Figure supplement 3.** Flow cytometry analysis of myeloid cells in IMQ-driven inflammation.

atypical monocytes (*Figure 3E and F* and *Figure 3—figure supplement 2A and B*; *Gamrekelashvili et al., 2016*), IMQ treatment induced the generation of F4/80$^{hi}$CD115$^+$ monocytes in the blood and increased MF in the spleen at d5 (*Figure 3E and F* and *Figure 3—figure supplement 3A and B*). This was followed by a peak in the DC population at d7 (*Figure 3E and F* and *Figure 3—figure supplement 3A and B*). These latter changes did not occur in bone marrow but were only observed in the periphery (*Figure 3—figure supplement 3C*). Together, these data suggest that wt Ly6C$^{hi}$ monocytes convert to Ly6C$^{lo}$ monocytes in response to TLR stimulation, while Notch2 deficient Ly6C$^{hi}$ monocytes differentiate into F4/80$^{hi}$CD115$^+$ monocytes, macrophages and DC, suggesting Notch2 as a master regulator of Ly6C$^{hi}$ monocyte cell fate during systemic inflammation.

## Global gene expression analysis identifies macrophage gene expression signatures in monocytes of Notch2-deficient mice during acute inflammation

To characterize more broadly the gene expression changes involved in monocyte differentiation during inflammation, we next subjected monocyte subsets from PB of wt and $N2^{\Delta My}$ mice after Sham or IMQ treatment (*Figure 4—figure supplement 1A*) to RNA-sequencing and gene expression analysis. After variance filtering and hierarchical clustering, 600 genes were differentially expressed between six experimental groups (*Figure 4A* and *Figure 4—source data 1*).

Principal component analysis (PCA) of differentially expressed genes (DEG) of all experimental groups revealed a clear separation between control Ly6C$^{hi}$ monocytes and IMQ-treated Ly6C$^{hi}$ or Ly6C$^{lo}$ monocytes. Interestingly, the effects of *Notch2* loss-of-function were most pronounced in the Ly6C$^{lo}$ populations, which separated quite strongly depending on genotype, while Ly6C$^{hi}$ monocytes from wt and $N2^{\Delta My}$ mice over all maintained close clustering under Sham or IMQ conditions (*Figure 4B and C* and *Figure 4—source data 1*).

Furthermore, while wt Ly6C$^{lo}$ cells were enriched for genes characteristic of patrolling monocytes (*Hes1, Nr4a1, Ace, Cd274 and Itgb3*), cells in the Ly6C$^{lo}$ gate from $N2^{\Delta My}$ mice showed upregulation of genes characteristic of mature phagocytes, such as MF (*Fcgr1, Mertk, C1qa, Clec7a, Maf, Cd36, Cd14, Adgre1* (encoding F4/80)) (*Figure 4D–F*).

Comparative gene expression analysis of Ly6C$^{lo}$ cell subsets during IMQ treatment identified 373 genes significantly up- or down-regulated with Notch2 loss-of-function (p-value<0.01, *Figure 4C–F* and *Figure 4—source data 2*), which were enriched for phagosome formation, complement system components, Th1 and Th2 activation pathways and dendritic cell maturation by ingenuity canonical pathway analysis (*Figure 4—figure supplement 1B*). Notably, signatures for autoimmune disease processes were also enriched (*Supplementary file 2*). Independent gene set enrichment analysis (GSEA) (*Isakoff et al., 2005*; *Mootha et al., 2003*) confirmed consistent up-regulation of gene sets in $N2^{\Delta My}$ Ly6C$^{lo}$ cells involved in several gene ontology biological processes, such as vesicle-mediated transport (GO:0016192), defense response (GO:0006952), inflammatory response (GO:0006954), response to bacterium (GO:0009617) and endocytosis (GO:0006897) (*Supplementary file 3* and *Figure 4—source data 2*). Overall, these data suggest regulation of Ly6C$^{hi}$ monocyte cell fate and inflammatory responses by Notch2.

Furthermore, changes in cell populations resulted in altered systemic inflammatory response patterns. Levels of TLR-induced cytokines and chemokines, such as TNF-α, CXCL1, IL-1β, IFN-α, were elevated to the same extend in wt and $N2^{\Delta My}$ mice in response to IMQ treatment, suggesting normal primary TLR-activation (*Figure 4—figure supplement 1C*). However, circulating levels of chemokines produced by Ly6C$^{lo}$ monocytes (*Carlin et al., 2013*), such as CCL2, CCL3, CXCL10, and IL-10

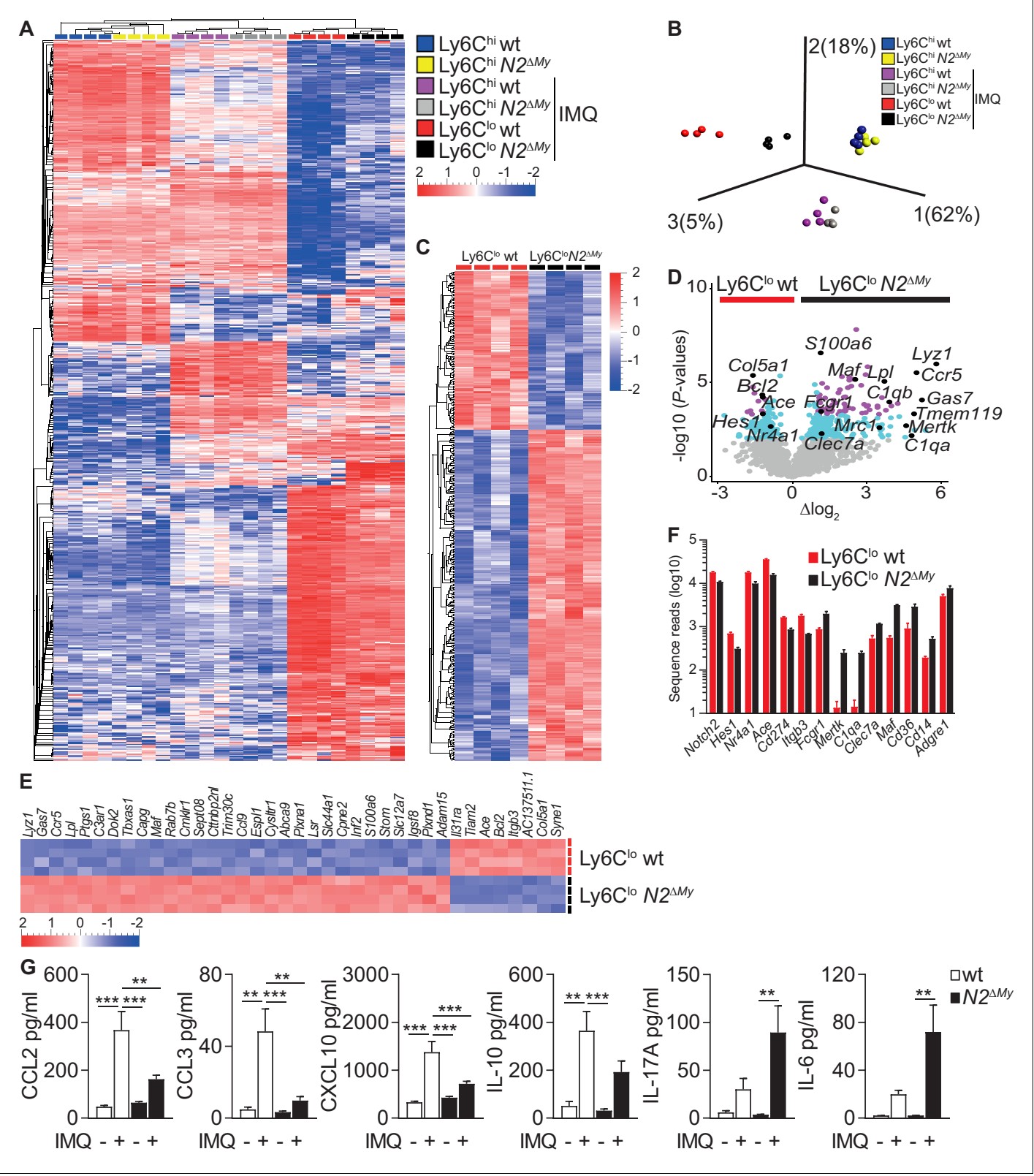

**Figure 4.** Enhanced macrophage gene expression signatures in monocytes and altered inflammatory response in $N2^{\Delta My}$ mice. (**A, B**) Hierarchical clustering of 600 ANOVA-selected DEGs (**A**) and PCA of PB monocyte subsets (**B**) after IMQ treatment (n = 4) is shown (Variance filtering 0.295, ANOVA followed by the B-H correction (p<0.0076, FDR ≤ 0.01)). (**C**) Hierarchical clustering of 373 DEGs from IMQ-treated wt and $N2^{\Delta My}$ Ly6C$^{lo}$ monocyte subsets (Variance filtering 0.117,–0.6≥Δlog₂ ≥ 0.6, Student's *t*-test with B-H correction (p<0.01, FDR ≤ 0.05)). (**D**) Volcano plot showing 379 DEGs

*Figure 4 continued on next page*

Figure 4 continued

between wt and $N2^{\Delta My}$ Ly6C$^{lo}$ cells (-log10(p-value) $\geq$ 2, FDR $\leq$ 0.05, light blue); and 87 DEGs ($-1\geq\Delta log_2 \geq 1$ (FDR $\leq$ 0.01) purple); genes of interest are marked black (Student's $t$-test with B-H correction). (E) Heat map of top 38 DEGs from (D) log10( p-value) $\geq$ 4,$-1\geq\Delta log_2 \geq 1$, Student's $t$-test with B-H correction. (F) Bar graph showing mean number and SEM of sequence reads for selected genes from IMQ-treated wt and $N2^{\Delta My}$ Ly6C$^{lo}$ cell subsets. (G) Analysis of cytokine and chemokine profiles in the serum of IMQ-treated mice. n = 5–10, pooled from four independent experiments. (G) *p<0.05, **p<0.01, ***p<0.001; 2way ANOVA with Bonferroni's multiple comparison test.

The online version of this article includes the following source data and figure supplement(s) for figure 4:

**Source data 1.** List of 600 DEGs for hierarchical clustering and PCA (*Figure 4A and B*) from Ly6C$^{hi}$ and Ly6C$^{lo}$ subpopulations isolated from sham-or IMQ(Aldara)-treated wt or $N2^{\Delta My}$ mice.

**Source data 2.** List of 373 DEGs between Ly6C$^{lo}$ cells isolated from IMQ(Aldara)-treated wt or $N2^{\Delta My}$ mice and used for the analysis in *Figure 4C and D*, *Figure 5A* and *Supplementary file 2–5*.

**Figure supplement 1.** Gating strategy for cell sorting, IPA and cytokine and chemokine analysis in IMQ-treated mice.

were higher in wt mice compared to $N2^{\Delta My}$ mice, while the levels of pro-inflammatory cytokines IL-17A, IL-6 and GMCSF were significantly enhanced in $N2^{\Delta My}$ mice but not in wt mice as compared to untreated controls, confirming systemic alterations in addition to cellular changes in *Notch2* loss-of-function mice in response to IMQ (*Figure 4G* and *Figure 4—figure supplement 1C*).

## Notch2-deficiency promotes macrophage differentiation

To match the observed gene expression pattern of inflammatory Ly6C$^{lo}$ cells from wt and $N2^{\Delta My}$ mice under IMQ treatment with previously described cells of the monocyte-macrophage lineage we performed pairwise gene set enrichment analysis with defined myeloid cell transcriptomic signatures using the GSEA software and BubbleGUM stand-alone software (*Isakoff et al., 2005*; *Mootha et al., 2003*; *Spinelli et al., 2015*; *Figure 5A* and *Supplementary file 4* and *5* and *Figure 4—source data 2*). Out of 29 transcriptomic signatures - representing tissue MF, monocyte derived DC (MoDC), conventional DC (cDC), plasmacytoid DC (pDC), classical (Ly6C$^{hi}$) monocytes (cMonocyte), non-classical (Ly6C$^{lo}$) monocytes (ncMonocyte) and B cells (as a reference) – significant enrichment was registered in seven signatures (normalized enrichment score (NES >1.5, FDR < 0.1)). The cell fingerprint representing ncMonocyte (#3) was highly enriched in the gene set from wt Ly6C$^{lo}$ monocytes (NES = 1.91, FDR = 0.039), while all other cell fingerprints showed no significant similarity (*Figure 5A* and *Supplementary file 4*), confirming a strong developmental restriction toward Ly6C$^{lo}$ monocytes in wt cells. In contrast, $N2^{\Delta My}$ gene sets showed the highest similarity (NES >1.8 and FDR < 0.01) with four cell fingerprints (#1, 5, 6, 7) representing different MF populations, and weak similarity to MoDC (NES = 1.64, FDR < 0.1) and cMonocyte (NES = 1.54, FDR < 0.1) (*Figure 5A* and *Supplementary files 4* and *5*). Phenotyping of cell populations by flow cytometry using MF markers MerTK and CD64 (*Figure 5B–D*) confirmed selective expansion of an F4/80$^{hi}$MerTK$^+$ (FM$^+$) monocyte population in IMQ-treated $N2^{\Delta My}$ mice (*Figure 5A* and *Supplementary files 4* and *5*). Together, these data demonstrate a cell fate switch from Ly6C$^{lo}$ monocytes toward macrophage signatures in the absence of Notch2.

## Notch2 regulates monocyte cell fate decisions during inflammation

In the steady-state, Ly6C$^{hi}$ monocytes differentiate into Ly6C$^{lo}$ monocytes and this process is regulated by Notch2 (*Gamrekelashvili et al., 2016*). In order to confirm that Notch2 controls differentiation potential of Ly6C$^{hi}$ monocytes in response to TLR stimulation, we performed adoptive transfer of CD45.2$^+$ wt or $N2^{\Delta My}$ BM Ly6C$^{hi}$ monocytes into IMQ-treated CD45.1$^+$ congenic recipients and analyzed the fate of donor cells after 3 days (*Figure 5—figure supplement 1A*). Unsupervised t-SNE analysis of flow cytometry data showed an expanded spectrum of expression patterns in cells from $N2^{\Delta My}$ donors compared to wt controls (*Figure 5—figure supplement 1B*). More precisely, Ly6C$^{hi}$ monocytes from wt mice converted preferentially to Ly6C$^{lo}$ monocytes (Ly6C$^{lo}$F4/80$^{lo/-}$CD11c$^+$-CD43$^+$MHC-II$^{lo/-}$ phenotype) during IMQ treatment (*Figure 5E and F* and *Figure 5—figure supplement 1C*). In contrast, conversion of Notch2-deficient Ly6C$^{hi}$ to Ly6C$^{lo}$ monocytes was strongly impaired, but the development of donor-derived F4/80$^{hi}$ macrophages in the spleen was strongly enhanced (*Figure 5G*). Furthermore, expansion of macrophages was also observed in aortas of Notch2 deficient mice in vivo after IMQ treatment (*Figure 5H*). Adoptive transfer studies confirmed that MF in IMQ-treated aortas originated from $N2^{\Delta My}$ Ly6C$^{hi}$ monocytes (*Figure 5J and K*).

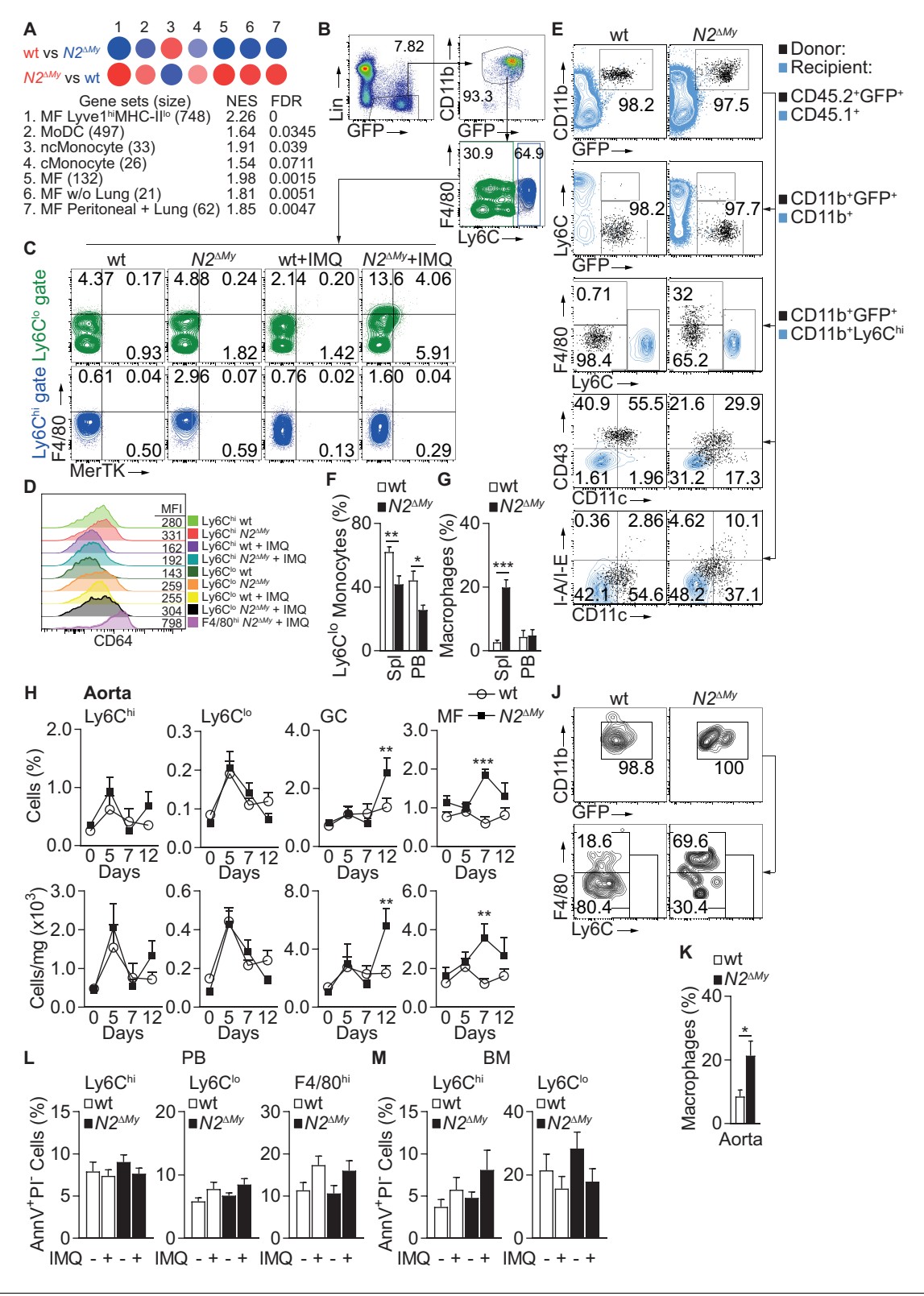

**Figure 5.** Notch2-deficient Ly6C[lo] cells show enhanced macrophage maturation during acute inflammation. (**A**) GSEA based on 373 DEGs between IMQ-treated wt and $N2^{\Delta My}$ Ly6C[lo] subsets in PB. Red – positive-, and blue - negative enrichment in corresponding color-coded wt or $N2^{\Delta My}$ cells. Size of the circle corresponds to NES and intensity of the color to FDR. (**B, C**) Representative flow cytometry plots showing expression of F4/80 and MerTK in gated Lin[-]CD45[+]CD11b[+]GFP[+]Ly6C[hi] and Lin[-]CD45[+]CD11b[+]GFP[+]Ly6C[lo] cells from PB of Sham- or IMQ-treated wt or $N2^{\Delta My}$ mice. (**D**) Representative

*Figure 5 continued on next page*

*Figure 5 continued*

flow cytometry histograms with corresponding mean fluorescence intensities (MFI) showing expression of CD64 on myeloid cells in PB of sham or IMQ-treated mice. (E) Flow cytometry analysis 3 days after adoptive transfer of wt or $N2^{\Delta My}$ BM CD45.2$^+$ Ly6C$^{hi}$ monocytes in IMQ-treated CD45.1$^+$ recipients. Transferred cells are shown in black and recipient CD45.1$^+$ (1$^{st}$ row), CD45.1$^+$CD11b$^+$ (2$^{nd}$ row) or CD45.1$^+$CD11b$^+$Ly6C$^{hi}$ cells (3$^{rd}$ −5$^{th}$ rows) are depicted in blue. (F, G) Frequency of donor-derived Ly6C$^{lo}$ monocytes (G) or macrophages (H) in CD45.2$^+$CD11b$^+$GFP$^+$ donor cells after adoptive transfer of wt or $N2^{\Delta My}$ Ly6C$^{hi}$ monocytes is shown. Data are pooled from three independent experiments (n = 6/9). (H) Relative (top) and absolute number (bottom) of myeloid subpopulations in aortas of untreated or IMQ-treated mice. Data are pooled from six experiments (n = 7–18). (J, K) Representative flow cytometry plot of donor CD11b$^+$GFP$^+$ cells (J) and relative frequency of donor-derived macrophages (K) within CD45.2$^+$CD11b$^+$GFP$^+$ cells recovered from aortas after adoptive transfer of wt or $N2^{\Delta My}$ Ly6C$^{hi}$ monocytes. (K) Data are pooled from three independent experiments (n = 9). (L, M) Relative frequency of apoptotic (AnnexinV$^+$PI$^{neg}$) cells in each myeloid subpopulation isolated from PB or BM of Sham- or IMQ-treated mice is shown (Data are from two independent experiments (n = 6–7)). (F, G, K) *p<0.05, **p<0.01, ***p<0.001; Student's *t*-test. (H, L, M) **p<0.01, ***p<0.001; two-way ANOVA with Bonferroni's multiple comparison test.

The online version of this article includes the following figure supplement(s) for figure 5:

**Figure supplement 1.** Characterization of F4/80$^{hi}$ monocytes in IMQ-treated mice.
**Figure supplement 2.** Expression of Notch2 and Notch-regulated gene in monocytes of IMQ-treated mice.

To evaluate Notch targeting efficiency in our system we performed flow cytometry and transcriptome analysis of transgenic monocyte subsets. In Ly6C$^{hi}$ monocytes, the *Lyz2$^{Cre}$*-mediated conditional deletion strategy induced a 40–50% reduction of Notch2 expression by flow cytometry and transcriptome analysis (*Figure 5—figure supplement 2A and C*), and a > 50% reduction of Notch target gene *Hes1*, demonstrating partial targeting and functional impairment of Notch2 in Ly6C$^{hi}$ monocytes (*Figure 5—figure supplement 2B*). At the same time, in Ly6C$^{lo}$ monocytes at baseline there was only minor reduction of Notch2, corroborating earlier results (*Gamrekelashvili et al., 2016*), but more efficient reduction of Notch2 in IMQ-treated mice to levels seen in Ly6C$^{hi}$ monocytes (*Figure 5—figure supplement 2C*). Furthermore, levels of Notch target gene *Hes1* were low in Ly6C$^{hi}$ monocytes but significantly increased in Ly6C$^{lo}$ monocytes (*Figure 5—figure supplement 2B*; *Gamrekelashvili et al., 2016*), suggesting low Notch signaling in Ly6C$^{hi}$ monocytes but higher Notch signaling activity in Ly6C$^{lo}$ monocytes.

Since Ly6C$^{lo}$ monocyte deficiency in *Nr4a1*-deficient mice is caused by increased apoptosis (*Hanna et al., 2011*), we analyzed cells by AnnexinV staining. Notch2-deficiency did not significantly increase cell death in all analyzed cell populations, neither during steady state nor during inflammation (*Figure 5L and M*).

Together, these data confirm that Notch2 is a master regulator of Ly6C$^{hi}$ monocyte differentiation potential, regulating a switch between Ly6C$^{lo}$ monocyte or macrophage cell fate during inflammation. These data also demonstrate that in the context of inactive myeloid Notch2 signaling, TLR-stimulation results in systemic pro-inflammatory changes and vascular inflammation.

## Discussion

Together, our data present a spectrum of developmental cell fates of Ly6C$^{hi}$ monocytes and their coordinated regulation by TLR and Notch signaling during inflammation. TLR and Notch signaling act independently and synergistically in promoting Ly6C$^{lo}$ monocyte development from Ly6C$^{hi}$ monocytes, while impairment in either signaling axis impairs Ly6C$^{lo}$ monocyte development. On the other hand, TLR stimulation in the absence of functional Notch2 signaling promotes macrophage gene expression signatures in monocytes and development of MF in aorta and spleen, suggesting Notch2 as a master regulator of Ly6C$^{hi}$ monocyte cell fate during systemic inflammation.

Plasticity of Ly6C$^{hi}$ monocytes ensures adaptation to environmental signals, which trigger distinct cell developmental programs inducing context- or tissue-specific subsets of terminally differentiated phagocytes, including Ly6C$^{lo}$ monocytes, MF or DC (*Guilliams et al., 2018*). In the steady-state, a subset of Ly6C$^{hi}$ monocytes converts to Ly6C$^{lo}$ monocytes in mice and humans, which is regulated by Notch2 and the endothelial Notch ligand Delta-like 1 (Dll1) (*Gamrekelashvili et al., 2016*; *Patel et al., 2017*; *Yona et al., 2013*). However, when recruited into tissues, Ly6C$^{hi}$ monocytes can give rise to two types of monocyte-derived resident tissue macrophages (MRTM) (*Chakarov et al., 2019*). These Lyve1$^{hi}$MHC-II$^{lo}$ and Lyve1$^{lo}$MHC-II$^{hi}$ MRTMs differ in phenotype and function as well as spatial distribution. Our gene set enrichment analysis revealed that the signature of cells from

inflamed wt mice showed highest and selective similarity to non-classical Ly6C$^{lo}$ monocytes, while *Notch2* loss-of-function cells showed the highest similarity to the gene set of Lyve1$^{hi}$MHC-II$^{lo}$ MRTM, but also a more general similarity to an extended spectrum of different MF signatures, suggesting Notch2 as a gate keeper of Ly6C$^{lo}$ monocytes vs. macrophage differentiation during inflammation. Due to the low number Ly6C$^{lo}$ monocytes at the steady state, we were not able to compare baseline expression profiles, which is a limitation of this analysis. Lyve1$^{hi}$MHC-II$^{lo}$ interstitial MFs are closely associated with blood vessels across different tissues and mediate inflammatory reactions (*Chakarov et al., 2019*). In line with this, *Notch2* knock-out mice showed a population of FM$^{+}$ monocytes with partial gene expression signatures of MF circulating in the blood and increased MF in the aorta, and adoptive transfer studies of Ly6C$^{hi}$ monocytes successfully recapitulated their differentiation into aortic MF. These data also have implications for the potential developmental regulation of MRTMs by TLR and Notch2. At the same time, the Notch2-deficient population showed weaker but significant enrichment of MoDC signatures, suggesting a mixture of cell subsets representing different stages of monocyte differentiation (*Menezes et al., 2016*; *Mildner et al., 2017*) or lineage commitment (*Liu et al., 2019*; *Yanez et al., 2017*) within this cell pool, although formally we cannot exclude progenitor contamination as a confounder in our adoptive cell transfer studies.

The *Lyz2$^{Cre}$*-mediated conditional deletion strategy induced a 40–50% reduction of Notch2 expression, and a > 50% reduction of Notch target gene *Hes1* in Ly6C$^{hi}$ monocytes, suggesting sufficient targeting and functional impairment of Notch2. Nevertheless, partial targeting might explain the small differences seen in PCA analysis in these cells. However, in light that baseline Notch signaling activity seems to be low in Ly6C$^{hi}$ monocytes and only significantly increases in Ly6C$^{lo}$ monocytes (*Gamrekelashvili et al., 2016*), it suggests that Notch2 influences cell fate decision in Ly6C$^{hi}$ monocytes or at the early stages of conversion to Ly6C$^{lo}$ monocytes. Furthermore, the fact that there is only a minor reduction of Notch2 in Ly6C$^{lo}$ monocytes at baseline, which suggests a strong selection bias against *Notch2* loss-of-function, further argues for a strict Notch2-dependence of monocyte conversion.

In the case of *Nr4a1* loss-of-function, the reduced numbers of Ly6C$^{lo}$ monocytes are due to increased apoptosis (*Hanna et al., 2011*). Although we did not find evidence for increased apoptosis due to Notch2-deficiency, our data do not exclude the possibility that regulation of cell survival by Notch2 contributes to the observed phenotype. In fact, two lines of the evidence suggest that regulation of cell survival might act synergistically to cell fate choices: first, in absolute numbers, there is no compensation by alternative cell fates for the number of lacking Ly6C$^{lo}$ monocytes in the blood or spleen of *N2$^{\Delta My}$* mice; second, expression of *Bcl2*, a strong regulator of cell survival, is downregulated in IMQ-treated *N2$^{\Delta My}$* Ly6C$^{lo}$ cells as compared to controls.

While our current data clearly demonstrate that *Notch2* loss-of-function promotes macrophage gene expression profiles in monocytes and macrophage development from Ly6C$^{hi}$ monocytes during TLR stimulation, we have previously shown that Dll1-Notch signaling promotes maturation of anti-inflammatory macrophages from Ly6C$^{hi}$ monocytes in ischemic muscle (*Krishnasamy et al., 2017*). Furthermore, Dll4-Notch signaling initiated in the liver niche was recently shown to promote Kupffer cell development after injury (*Bonnardel et al., 2019*; *Sakai et al., 2019*) or to promote pro-inflammatory macrophage development (*Xu et al., 2012*). This suggests that the role of Notch is ligand-, cell- and context-specific, which emphasizes the differential effects of specific ligand-receptor combinations (*Benedito et al., 2009*). Our data demonstrate that Notch2 is a master regulator of Ly6C$^{hi}$ monocyte cell fate during inflammation, which contributes to the nature of the inflammatory response.

Lastly, our data also reveal a potentially important function of myeloid Notch2 for regulation of systemic and vascular inflammation with implications for autoimmune disease. When wt mice are challenged with TLR stimulation they show predominant conversion of Ly6C$^{hi}$ monocytes into Ly6C$^{lo}$ monocytes with blood vessel patrolling and repairing function (*Carlin et al., 2013*) and IL-10 secretion. In contrast, *Notch2* knock-out mice show predominant and ectopic differentiation of Ly6C$^{hi}$ monocytes into FM$^{+}$ monocytes and macrophages, which appear in the bloodstream and the spleen and infiltrate major blood vessels, such as the aorta, along with aberrant cytokine profiles. In addition, absence of functional Notch2 promoted a core macrophage signature and strong upregulation of canonical pathways involved in autoimmune disease. Since TLR7 has been implicated in the development of autoimmune disease (*Santiago-Raber et al., 2011*; *Santiago-Raber et al., 2010*), our data suggest Notch2 as an important modulator of this process by regulating cell differentiation and

systemic inflammation. However, the relevance and possible disease context requires further studies.

## Materials and methods

### Mice

B6.129P-*Cx3cr1*[tm1Litt]/J (*Cx3cr1*[GFP/+]) mice (*Jung et al., 2000*), B6.129P2-*Lyz2*[tm1(cre)Ifo] (*Lyz2*[Cre]) mice (*Clausen et al., 1999*), B6.129-*Notch2*[tm1Frad]/J (*Notch2*[lox/lox]) mice (*Besseyrias et al., 2007*), B6.129-*Lyz2*[tm1(cre)Ifo]*Notch2*[tm1Frad]*Cx3cr1*[tm1Litt] (*N2*[ΔMy]) (*Gamrekelashvili et al., 2016* have been previously described. B6.SJL-*Ptprc*[a]*Pepc*[b]/BoyJ (CD45.1[+]) mice were from central animal facility of Hannover Medical School (ZTL, MHH). B6.129P2-*Myd88*[tm1Hlz]/J (*Myd88*[-/-]) (*Gais et al., 2012*) and *Myd88*[+/+] littermate control (wt) mice were kindly provided by Dr. Matthias Lochner. Mice were housed under specific pathogen-free conditions in the animal facility of Hannover Medical School.

### Tissue and cell preparation

For single cell suspension mice were sacrificed and spleen, bone marrow, blood and aortas were collected. Erythrocytes were removed by red blood cell lysis buffer (BioLegend) or by density gradient centrifugation using Histopaque 1083 (Sigma-Aldrich). Aortas were digested in DMEM medium supplemented with 500 U/ml Collagenase II (Worthington). After extensive washing, cells were resuspended in PBS containing 10%FCS and 2 mM EDTA kept on ice, stained and used for flow cytometry or for sorting.

### Flow cytometry and cell sorting

Non-specific binding of antibodies to Fc-receptors was blocked with anti-mouse CD16/CD32 (TruStain fcX from BioLegend) in single-cell suspensions prepared from Spl, PB or BM. After subsequent washing step, cells were labeled with primary and secondary antibodies or streptavidin-fluorochrome conjugates and were used for flow cytometry analysis (LSR-II, BD Biosciences) or sorting (FACSAria; BD Biosciences or MoFlo XDP; Beckman Coulter).

For apoptosis assay, single-cell suspensions were stained with primary and secondary antibodies, washed, re-suspended in AnnexinV binding buffer (Biolegend) and transferred into tubes. Cells were stained with AnnexinV (AnnV) and propidium iodide (PI) at room temperature for 20 min and were immediately analyzed by flow cytometry. Antibodies and fluorochromes used for flow cytometry are described in *Supplementary file 6*. Flow cytometry data were analyzed using FlowJo software (FlowJo LLC). Initially cells were identified based on FSC and SSC characteristics. After exclusion of doublets (on the basis of SSC-W, SSC-A), relative frequency of each subpopulation from live cell gate, or absolute number of each subset (calculated from live cell gate and normalized per Spl, per mg Spl, mg BM, mg aorta or μl PB) were determined and are shown in the graphs as mean ± SEM, unless otherwise stated. Unsupervised t-distributed stochastic neighbor embedding (t-SNE) analysis (*van der Maaten and Hinton, 2008*) was performed on live CD45[+]Lin[-]GFP[+]CD11b[+] population in concatenated samples using FlowJo.

### Cytokine multiplex bead-based assay

Sera were collected from control or Aldara treated mice and kept frozen at −80°C. Concentration of IFN-γ, CXCL1, TNF-α, CCL2, IL-12(p70), CCL5, IL-1β, CXCL10, GM-CSF, IL-10, IFN-β, IFN-α, IL-6, IL-1α, IL-23, CCL3, IL-17A were measured with LEGENDplex multi-analyte flow assay kits (BioLegend) according to manufacturer's protocol on LSR-II flow cytometer. Data were processed and analyzed with LEGENDplex data analysis software (BioLegend).

### RNA isolation, library construction, sequencing and analysis

Peripheral blood monocyte subpopulations were sorted from Aldara treated mice or untreated controls (*Figure 5—figure supplement 1A*) and RNA was isolated using RNeasy micro kit (Qiagen). Libraries were constructed from total RNA with the 'SMARTer Stranded Total RNA-Seq Kit v2 – Pico Input Mammalian' (Takara/Clontech) according to manufacturer's recommendations, barcoded by dual indexing approach and amplified with 11 cycles of PCR. Fragment length distribution was monitored using 'Bioanalyzer High Sensitivity DNA Assay' (Agilent Technologies) and quantified by 'Qubit

dsDNA HS Assay Kit' (ThermoFisher Scientific). Equal molar amounts of libraries were pooled, denatured with NaOH, diluted to 1.5pM (according to the Denature and Dilute Libraries Guide (Illumina)) and loaded on an Illumina NextSeq 550 sequencer for sequencing using a High Output Flowcell for 75 bp single reads (Illumina). Obtained BCL files were converted to FASTQ files with bcl2fastq Conversion Software version v2.20.0.422 (Illumina). Pre-processing of FASTQ inputs, alignment of the reads and quality control was conducted by nfcore/rnaseq (version 1.5dev) analysis pipeline (The National Genomics Infrastructure, SciLifeLab Stockholm, Sweden) using Nextflow tool. The genome reference and annotation data were taken from GENCODE.org (GRCm38.p6 release M17). Data were normalized with DESeq2 (Galaxy Tool Version 2.11.39) with default settings and output counts were used for further analysis with Qlucore Omics explorer (Sweden). Data were $\log_2$ transformed, 1.1 was used as a threshold and low expression genes (<50 reads in all samples) were removed from the analysis. Hierarchical clustering (HC) or principal component analysis (PCA) was performed on 600 differential expressed genes (DEGs) after variance filtering (filtering threshold 0.295) selected by ANOVA with the Benjamini-Hochberg (B-H) correction (p<0.01, FDR $\leq$ 0.01). For two group comparisons Student's $t$-test with B-H correction was used and 373 DEGs ((Variance filtering 0.117,–$0.6 \geq \Delta \log_2 \geq 0.6$ (p<0.01, FDR $\leq$ 0.05)) were selected for further IPA or GSEA analysis.

Ingenuity pathway analysis (IPA) was performed on 373 DEGs using IPA software (Qiagen) with default parameters. Top 20 canonical pathways and top five immunological diseases enriched in DEGs were selected for display.

Gene set enrichment analysis (GSEA) (*Mootha et al., 2003*; *Subramanian et al., 2005*) was performed on 373 DEGs using GSEA software (Broad institute) and C5 GO biological process gene sets (*Liberzon et al., 2015*) from MsigDB with 1000 gene set permutations for computing p-values and FDR.

BubbleGUM software, an extension of GSEA (*Spinelli et al., 2015*; *Vu Manh et al., 2015*), GSEA software (Broad Institute) and published transcriptomic signatures (*Chakarov et al., 2019*; *Gautier et al., 2012*; *Schlitzer et al., 2015*; *Vu Manh et al., 2015*) were used to assess and visualize the enrichment of obtained gene sets for myeloid populations and define the nature of the cells from which the transcriptomes were generated.

## In vitro conversion studies

96-well flat bottom plates were coated at room temperature for 3 hr with IgG-Fc or DLL1-Fc ligands (all from R and D) reconstituted in PBS. Sorted BM Ly6C$^{hi}$ monocytes were cultured in coated plates and were stimulated with Resiquimod (R848, 0.2 µg ml$^{-1}$, Cayman Chemicals) or LPS (0.2 µg ml$^{-1}$, *E. coli* O55:B5 Sigma-Aldrich) in the presence of M-CSF (10 ng ml$^{-1}$, Peprotech) at 37°C for 24 hr. One day after culture, cells were harvested, stained and subjected to flow cytometry. Relative frequency (from total live CD11b$^+$GFP$^+$ cells) or absolute numbers of Ly6C$^{lo}$ monocyte-like cells (CD11b$^+$GFP$^+$Ly6C$^{lo/-}$CD11c$^{lo}$MHC-II$^{lo/-}$CD43$^+$) recovered from each well served as an indicator of conversion efficiency and is shown in the graphs. Alternatively, cultured cells were harvested and isolated RNA was used for gene expression analysis.

## Induction of acute systemic inflammation using IMQ

Mice were anesthetized and back skin was shaved and depilated using depilating crème. Two days after 50 mg/mouse/day Aldara (containing 5% Imiquimod, from Meda) or Sham crème were applied on depilated skin and right ear (where indicated) for 4–5 consecutive days (*El Malki et al., 2013*; *van der Fits et al., 2009*). Mouse weight and ear thickness were monitored daily. Mice were euthanized on the indicated time points after start of treatment (day 0, 5, 7 and 12) PB, Spl, BM and aortas were collected for further analysis.

## Adoptive cell transfer experiments

Lin$^-$CD11b$^+$Ly6C$^{hi}$GFP$^+$ monocytes were sorted from BM of CD45.2$^+$ donors and injected into CD45.1$^+$ recipients intravenously (i.v.). In separate experiment wt or *Myd88$^{-/-}$* LinCD45.2$^+$CD11b$^+$-Ly6C$^{hi}$CX$_3$CR1$^+$ monocytes were used as a donor for transfer. 30 min later PBS or R848 (37.5 µg per mouse) were injected in recipient mice. Two days after transfer Spl, PB and BM were collected and single cell suspension was prepared. After blocking of Fc receptors (anti-mouse CD16/CD32, TruStain fcX from BioLegend), cells were labeled with biotin-conjugated antibody cocktail containing anti-

CD45.1 and anti-Lin (anti-CD3, CD19, B220, NK1.1, Ly6G, Ter119) antibodies, anti-biotin magnetic beads and enriched on LS columns (Miltenyi Biotec) according to manufacturer's instructions. CD45.1$^{neg}$Lin$^{neg}$ fraction was collected, stained and analyzed by flow cytometry. Ly6C$^{lo}$ monocytes (CD45.2$^+$CD11b$^+$GFP$^+$Ly6C$^{lo/-}$F4/80$^{lo}$CD11c$^{lo}$CD43$^{hi}$ cells) and macrophages (CD45.2$^+$CD11b$^+$-GFP$^+$Ly6C$^{lo/-}$F4/80$^{hi}$CD115$^+$ cells) were quantified in Spl, BM and PB as relative frequency of total donor derived CD45.2$^+$CD11b$^+$GFP$^+$ cells. Adoptive transfer experiments in IMQ-treated mice were performed using wt or $N2^{\Delta My}$ Lin$^-$CD45.2$^+$CD11b$^+$Ly6C$^{hi}$GFP$^+$ donor monocytes and Spl, PB and aortas of recipients were analyzed 3 days after transfer.

## Quantitative real-time PCR analysis

Total RNA was purified from cell lysates using Nucleospin RNA II kit (Macherey Nagel). After purity and quality check, RNA was transcribed into cDNA employing cDNA synthesis kit (Invitrogen) according to manufacturer's instructions. Quantitative real-time PCR was performed using specific primers for *Nr4a1*: forward, 5'-AGCTTGGGTGTTGATGTTCC-3', reverse, 5'-AATGCGATTCTGCAGC TCTT-3' and *Pou2f2*: forward, 5'-TGCACATGGAGAAGGAAGTG-3', reverse, 5'-AGCTTGGGACAA TGGTAAGG-3' and FastStart Essential DNA Green Master on a LightCycler 96 system from Roche according to manufacturer's instructions. Expression of each specific gene was normalized to expression of *Rps9* and calculated by the comparative CT ($2^{-\Delta\Delta CT}$) method (*Schmittgen and Livak, 2008*).

## Statistical analysis

Results are expressed as mean ± standard error of mean (SEM). N numbers are biological replicates of experiments performed at least three times unless otherwise indicated. Significance of differences was calculated using unpaired, two-tailed Student's *t*-test with confidence interval of 95%. For comparison of multiple experimental groups one-way or two-way ANOVA and Bonferroni's multiple-comparison test was performed.

# Acknowledgements

We thank the Central Animal Facility, Research Core Facility Cell Sorting and Research Core Unit Genomics of Hannover Medical School for support. We thank Thien-Phong Vu Manh and Lionel Spinelli for consultation on BubbleGUM. Funded by grants from DFG (GA 2443/2–1), DSHF (F/17/16) to JG, and DFG (Li948-7/1) to FPL.

# Additional information

## Funding

| Funder | Grant reference number | Author |
| --- | --- | --- |
| Deutsche Forschungsge-meinschaft | GA 2443/2-1 | Jaba Gamrekelashvili |
| Deutsche Forschungsge-meinschaft | Li948-7/1 | Florian P Limbourg |
| Deutsche Stiftung für Herz-forschung | F/17/16 | Jaba Gamrekelashvili |

The funders had no role in study design, data collection and interpretation, or the decision to submit the work for publication.

## Author contributions

Jaba Gamrekelashvili, Conceptualization, Resources, Data curation, Formal analysis, Funding acquisition, Validation, Investigation, Visualization, Methodology, Writing - original draft, Project administration, Writing - review and editing; Tamar Kapanadze, Data curation, Writing - review and editing; Stefan Sablotny, Resources, Data curation; Corina Ratiu, Data curation, Formal analysis, Writing - review and editing; Khaled Dastagir, Andre Sitnow, Data curation; Matthias Lochner, Tim Sparwasser, Resources, Methodology, Writing - review and editing; Susanne Karbach, Philip Wenzel, Ulrich Kalinke, Bernhard Holzmann, Hermann Haller, Conceptualization, Resources, Writing - review

and editing; Susanne Fleig, Conceptualization, Data curation, Writing - review and editing; Florian P Limbourg, Conceptualization, Resources, Supervision, Funding acquisition, Validation, Investigation, Methodology, Project administration, Writing - review and editing

### Author ORCIDs
Jaba Gamrekelashvili (ID) https://orcid.org/0000-0001-7533-6906
Florian P Limbourg (ID) https://orcid.org/0000-0002-8313-7226

### Ethics
Animal experimentation: All experiments were performed with 8-12 weeks old mice and age and sex matched littermate controls with approval of the local animal welfare board LAVES (Niedersächsisches Landesamt für Verbraucherschutz und Lebensmittelsicherheit), Lower Saxony, Animal Studies Committee, animal study proposals #14-1666, #16-2251, #18-2777, #2014-63, #2018-221). Mice were housed in the central animal facility of Hannover Medical School (ZTL) and were maintained and supervised as approved by the Institutional Animal Welfare Officer (Tierschutzbeauftragter).

### Decision letter and Author response
Decision letter https://doi.org/10.7554/eLife.57007.sa1
Author response https://doi.org/10.7554/eLife.57007.sa2

## Additional files

### Supplementary files
• Supplementary file 1. Surface phenotype signatures for identification of distinct myeloid populations in vivo. Lin: CD3, CD45R/B220, CD19, NK1.1, Ly6G, Ter119.

• Supplementary file 2. IPA of top five immunological diseases enriched in Ly6C$^{lo}$ cells from IMQ-treated $N2^{\Delta My}$ mice.

• Supplementary file 3. Top 20 gene sets involved in GO biological processes enriched in Ly6C$^{lo}$ cells from IMQ-treated $N2^{\Delta My}$ mice.

• Supplementary file 4. Parameters and the results of GSEA performed on 373 DEGs for *Figure 5A*.

• Supplementary file 5. List of the genes enriched in Lyve1$^{hi}$MHC-II$^{lo}$ MF gene set from *Figure 5A*.

• Supplementary file 6. List of antibodies and fluorescence dyes for flow cytometry used in the study.

• Transparent reporting form

### Data availability
All data generated or analysed during this study are included in the manuscript and supporting files. Data from RNA sequencing have been deposited to NCBI's Gene Expression Omnibus and are available under the accession number GSE147492.

The following dataset was generated:

| Author(s) | Year | Dataset title | Dataset URL | Database and Identifier |
|---|---|---|---|---|
| Gamrekelashvili J, Limbourg FP | 2020 | Notch and TLR signaling coordinate monocyte cell fate and inflammation | https://www.ncbi.nlm.nih.gov/geo/query/acc.cgi?acc=GSE147492 | NCBI Gene Expression Omnibus, GSE147492 |

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
