## [Decision Letter]

**Acceptance summary:**

The authors have explored the processes involved in the fate of LY6C^hi^ monocytes and the factors influencing this process under TLR7-induced inflammation. They show that the Ly6C^hi^ monocyte conversion to Ly6C^lo^ monocytes is dependent on cell intrinsic Notch2 and Myd88 pathway as selective invalidation of Notch2 bias the differentiation of monocytes in steady state and more extensively in inflammatory condition toward circulating macrophages. This is an interesting set of findings, based on sound experiments using a variety of approaches ranging over conditional genetic deletions, cell transfers and cell culture techniques to track these cells in vivo and in vitro.

**Decision letter after peer review:**

Thank you for submitting your article "Notch and TLR signaling coordinate monocyte cell fate and inflammation" for consideration by *eLife*. Your article has been reviewed by Satyajit Rath as the Senior Editor, a Reviewing Editor, and three reviewers. The following individuals involved in review of your submission have agreed to reveal their identity: Alexandre Boissonnas (Reviewer #3).

The reviewers have discussed the reviews with one another and the Reviewing Editor has drafted this decision to help you prepare a revised submission.

As the editors have judged that your manuscript is of interest, but as described below that additional experiments are required before it is published, we would like to draw your attention to changes in our revision policy that we have made in response to COVID-19 (https://elifesciences.org/articles/57162). First, because many researchers have temporarily lost access to the labs, we will give authors as much time as they need to submit revised manuscripts. We are also offering, if you choose, to post the manuscript to bioRxiv (if it is not already there) along with this decision letter and a formal designation that the manuscript is 'in revision at *eLife*'. Please let us know if you would like to pursue this option. (If your work is more suitable for medRxiv, you will need to post the preprint yourself, as the mechanisms for us to do so are still in development.)

Summary:

The authors explored the processes involved in the fate of LY6C^hi^ monocytes and the factors influencing this process under TLR7-induced inflammation. They show that the Ly6C^hi^ monocyte conversion to Ly6C^lo^ monocytes is dependent on cell intrinsic Notch2 and Myd88 pathway as selective invalidation of Notch2 bias the differentiation of monocytes in steady state and more extensively in inflammatory condition toward circulating macrophages. This is an interesting paper and the presented experiments are of good quality. This study uses a variety of conditional genetic deletions, cell transfers and cell culture techniques to track these cells in vivo and in vitro.

Essential revisions:

While the editors and reviewers all agreed on the interest of the study, several points need to be addressed to strengthen the study.

The transcriptome analysis presented in Figure 4 is important, but missing Ly6Clow monocytes from wt and myeloid Notch deficient cells under steady state conditions. It would be important to compare the profiles of untreated and IMQ-treated Notch2-deficient cells to dissect the treatment effects from the steady state condition caused by Notch2 deletion.

The authors should provide deletion efficiency in both monocyte subsets, for instance by qRT-PCR for the deleted exon. There is experimental proof that the Cre lox system is not fully efficient in classical monocytes, usually associated to their short life span and likely partial in the ncMo (as confirmed in Gamrekelashvili et al., 2016) and others. This could explain why the PCA analysis detects minimal difference in the Ly6Chigh subset as well as no impact on numbers in Notch2-mutant mice. Hence it is dificult to support the main conclusion that Notch2-mediated decision occurs at the Ly6Chigh level as it is still mostly present. Notch2 could rather regulate the survival of Ly6Clow Mo. This point is slightly discussed in the Discussion section. The efficiency of Notch recombination in Mo subsets after the different treatments must be presented (even if previously published at steady state) to better apprehend this limit.

Lineage tracing is always challenging, and it is difficult to assess whether notch signaling is requested in the differentiation toward a specific lineage or is requested for cell survival. Notch regulated monocyte survival is a well taken point, since also Bcl2 is strongly decreased in Notch2-/- monocytes. Therefore, providing absolute numbers for the in vivo and in vitro experiments (cell numbers instead of %) is absolutely required. In fact, most of the study is based on % (often unclear among which population) which could lead to misinterpretation. The authors should provide absolute numbers per organs along with per mg, whenever possible. For example, "By comparison, the TLR4 ligand LPS also increased Ly6C^lo^ cell numbers and expression levels of Nr4a1and Pou2f2. However, the absolute conversion rate was lower under LPS and there was no synergy with DLL1 (Figure 1D and E)". The numbers are not evaluated here neither the conversion rate as long as survival difference cannot be excluded.

The 'unrestrained inflammation' part either should be experimentally proven our should be completely rephrased (or deleted). The authors state in the abstract that "the absence of functional Notch2 signaling promotes resident tissue macrophage gene expression (…) resulting in unrestrained systemic inflammation" that could be interpreted as an overstatement. The inflammatory response shows perturbation in Notch-deficient mice, but not a clear pro-inflammatory shift (see also point below). Accordingly, it is not clear, if the splenomegaly or the "unrestrained systemic inflammation" is directly caused by monocytes. LysM-Cre is also active in neutrophils, which similarly express high levels of Notch2 according to immgen and can contribute to the observed phenotype. If the authors want to keep the link of 'Notch2-deficient monocytes cause unrestrained systemic inflammation' then the authors should perform monocyte (anti-CCR2 treatment) and neutrophil (anti-Ly6G treatment) depletion experiments in IMQ-treated wt and Notch-deficient mice. If the observed splenomegaly in Notch2-deficient mice is reduced to wt levels when treated with CCR2 (but not after Ly6G treatment), it is likely that monocytes are the direct cause.

T0 purity and flow analysis (F4/80, Ly6C,CD43, CD11c CD11b and GFP) of the sorted monocyte from all mouse strains should be provided. Working with bone marrow monocytes can be precarious, as the bone marrow may be contaminated with progenitors (and should be mentioned in the text).

In Figure 3 the authors perform t-SNE analysis based on the Ly6C, CD43, MHCII, F4/80 and CD11c marker set and according to this identified 5 monocyte 'subsets' (Figure 2C). Please also show the corresponding flow cytometry analysis (dot plots) (especially for PB) to identify these 5 subsets by regular gating and to see intensity of especially F4/80 and MHCII staining in Ly6Chigh and Ly6Clow monocytes in all conditions. In addition, in the gating strategy Figure 3—figure supplement 1C the authors used F4/80 to discriminate CD115+ MF from monocytes. Rose et al., 2011 in Cytometry A showed that splenic monocytes also express F4/80 and that this antigen can be used to identify monocytes. Therefore, it is possible that MF cells in the authors' gating strategy are contaminated by (probably aged) Ly6C low monocytes that up-regulated F4/80. To counter argue this please show their gated MF in a FACS plot with Ly6C vs CD43.

Consistent with the working hypothesis: Notch deficient LY6C^lo^ monocyte phenotype drastically changes following IMQ Figure 3. Why were Ly6C^lo^ wt and N2 deficient monocytes not examined without IMQ Figure 4? It is important to know how these cells alter from baseline and help distinguish if the effects are IMQ alone, Notch alone or IMQ and Notch.

The conclusion from these studies is in the presence of IMQ and absence of Notch2 LY6C^hi^ cells become more of a "macrophage" compared to the natural progression towards LY6C^lo^ monocytes Figure 5A/B. In Figure 5B, c and F what is the phenotype without IMQ. At present, without the controls, it is hard to comment on these experiments.

[Editors' note: further revisions were suggested prior to acceptance, as described below.]

Thank you for submitting your article "Notch and TLR signaling coordinate monocyte cell fate and inflammation" for consideration by *eLife*. Your article has been reviewed by Satyajit Rath as the Senior Editor, a Reviewing Editor, and three reviewers. The following individuals involved in review of your submission have agreed to reveal their identity: Alexander Mildner (Reviewer #2); Alexandre Boissonnas (Reviewer #3).

The reviewers have discussed the reviews with one another and the Reviewing Editor has drafted this decision to help you prepare a revised submission.

Summary:

All reviewers and editor agree that the authors have addressed most of the concerns that were raised and included important new data to support their hypothesis. Even though the authors could not perform the important bulk RNA-seq experiment of Notch-deficient Ly6C low monocytes under steady state conditions, they represented (and included in the manuscript) strong reasons for this (especially the mentioned selection bias). However, one last point on survival needs to be addressed and discussed in the revised version of the manuscript.

Annexin V detection ex vivo cannot rule out the reduced survival of Ly6CClow monocyte

Apoptotic cells are not harvested for flow cytometry as the process is not synchronous. Hence, the hypothesis that Notch 2 control Mo survival should not be excluded. This alternative hypothesis is even supported by the absolute count where the number of missing Ly6Clow is not compensated by both MF and DC numbers in Notch 2 deficient in Figure 3. For instance, the difference of 200 cells/µl of Ly6chigh in the blood between WT and N2KO shows a difference of 50 cells /µl of Ly6C^lo^ at day 7 and a difference of 1.5 F4/80^hi^ cells /µl which rather argue for a defect in cell survival and a very weak accumulation of F4/80^hi^ cells in the blood. Similar discrepancy is observed in the spleen. Around 800,000 Ly6C^lo^ cell difference in the spleen for only an extra of 10,000 F4/80^hi^ cells.

Absolute value for adoptive transfer experiment have not been provided. In addition, since bcl2 is strongly down in ko cells, a reduced survival does not necessarily mean a higher apoptosis rate.

The authors need to discuss and state this survival point across the manuscript, whenever necessary.

---

## [Author Response]

Summary:The authors explored the processes involved in the fate of LY6C^hi^ monocytes and the factors influencing this process under TLR7-induced inflammation. They show that the Ly6C^hi^ monocyte conversion to Ly6C^lo^ monocytes is dependent on cell intrinsic Notch2 and Myd88 pathway as selective invalidation of Notch2 bias the differentiation of monocytes in steady state and more extensively in inflammatory condition toward circulating macrophages. This is an interesting paper and the presented experiments are of good quality. This study uses a variety of conditional genetic deletions, cell transfers and cell culture techniques to track these cells in vivo and in vitro.Essential revisions:1) While the editors and reviewers all agreed on the interest of the study, several points need to be addressed to strengthen the study.The transcriptome analysis presented in Figure 4 is important, but missing Ly6Clow monocytes from wt and myeloid Notch deficient cells under steady state conditions. It would be important to compare the profiles of untreated and IMQ-treated Notch2-deficient cells to dissect the treatment effects from the steady state condition caused by Notch2 deletion.

We agree that it is important to dissect the impact of Notch2 loss-of-function and IMQ treatment and that it would be desirable to also compare the Ly6C^lo^ baseline sets. However, there are technical/methodological issues that unfortunately prevented this. Due to the low numbers of circulating cells in mutant mice, we failed to get enough material for sequencing and could not include this condition in the analysis. More specifically, in two independent sorts, each time pooling three mutant mice, we retrieved only 8278 and 15300 Ly6C^lo^ cells, which are several orders below the required input of 100000 cells for sequencing. Also, there seems to be a strong selection bias in Ly6C^lo^ cells against Notch2 loss of function at steady state, based on Notch2 expression data by flow cytometry (see also point 2 below). As a consequence, while the original Ly6C^hi^ monocytes -as a source of Ly6C^lo^ development- show a 50% reduction in Notch2 expression, the few remaining Ly6C^lo^ cells in Notch2 mutant mice show roughly the same level of Notch2 expression as control mice (Figure 5—figure supplement 2C), arguing that only Notch2 competent cells complete conversion to Ly6C^lo^ cells at the steady state. A comparison might therefore not be representative of Ly6C^lo^ cells with Notch2 loss of function. We have included a discussion on the limitations of the study in the Discussion section (Due to the low number Ly6C^lo^ monocytes at the steady state we were not able to compare baseline expression profiles, which is a limitation of this analysis.)

While we are not able to provide baseline transcriptome analysis, we would like to point out that our cell phenotype and population analysis by flow cytometry clearly shows effects of Notch2 deletion and IMQ-treatment on monocytes, separately and in combination, including baseline data (Figure 3 and Figure 3—figure supplement 2). In particular, there is no evidence of a significant F4/80^hi^ monocyte population in Notch2 deficient mice under steady state conditions, but only with treatment of Notch2 knock-out mice, demonstrating that development of F4/80^hi^ monocytes and macrophages occurs only under inflammatory conditions in Notch2 deficient mice. Also, upregulation of macrophage markers Mertk and CD64 was not observed to a relevant degree in baseline Ly6C^lo^ cells from Notch deficient mice. Thus, a major cell type characteristic of the mutant response to IMQ is not observed under baseline conditions. This is supported by our previous findings that adoptive transfer of Notch2 deficient Ly6C^hi^ monocytes under steady state conditions (Gamrekelashvili et al., 2016) does not lead to development of F4/80^hi^ cells as observed in current study (Figure 5E).

2) The authors should provide deletion efficiency in both monocyte subsets, for instance by qRT-PCR for the deleted exon. There is experimental proof that the Cre lox system is not fully efficient in classical monocytes, usually associated to their short life span and likely partial in the ncMo (as confirmed in Gamrekelashvili et al., 2016) and others. This could explain why the PCA analysis detects minimal difference in the Ly6Chigh subset as well as no impact on numbers in Notch2-mutant mice. Hence it is dificult to support the main conclusion that Notch2-mediated decision occurs at the Ly6Chigh level as it is still mostly present. Notch2 could rather regulate the survival of Ly6Clow Mo. This point is slightly discussed in the Discussion section. The efficiency of Notch recombination in Mo subsets after the different treatments must be presented (even if previously published at steady state) to better apprehend this limit.

Again, we agree that data on deletion efficiency are important to put the presented data on Ly6C^hi^ and Ly6C^lo^ monocytes in perspective. We have analyzed deletion efficiency of Notch2 and Notch target gene expression by transcriptome analysis and Notch2 surface expression by flow cytometry, the data are summarized in Figure 5—figure supplement 2 and are reported in subsection “Notch2 regulates monocyte cell fate decisions during inflammation”.

With respect to Ly6C^hi^ monocytes, our new analysis in these cells shows that there is a 40-50% reduction of Notch2 expression by sequence reads and by surface staining (Figure 5—figure supplement 2A, C). Furthermore, there is > 50% reduction of Notch target gene Hes1, suggesting sufficient targeting and functional impairment of Notch2 in Ly6C^hi^ monocytes. In light that baseline Notch signaling activity seems to be low in Ly6C^hi^ monocytes and only significantly increases in Ly6C^lo^ monocytes (see Hes1 expression in Figure 5—figure supplement 2B and Gamrekelashvili, 2016, Figure. 2A) this is consistent with an interpretation that Notch2 impairment does not significantly influence Ly6C^hi^ gene expression or numbers, since it seems to be not particularly active. It also supports the argument that while Ly6C^hi^ cells per se do not depend on fully functional Notch2 signaling, conversion to Ly6C^lo^ cells depends on Notch2, which might happen at the level of Ly6C^hi^ cells or below. However, we also agree that partial targeting of Notch2 is an alternative explanation for the small differences seen in PCA analysis. We have clarified this in the Discussion section.

With respect to Ly6C^lo^ cells, there is minor reduction of Notch2 in the few remaining mutant cells at baseline, but more efficient reduction of Notch2 in IMQ mice, to levels seen in Ly6C^hi^ cells (Figure 5—figure supplement 2C). This would be in line with selection pressure against loss-of-function (see above) at baseline, either because of inability to initiate conversion (in line with our data on adoptive transfer of Ly6C^hi^ monocytes, supporting that decisions that place at the level of Ly6C^hi^ cells) or enhanced cell death in Ly6C^lo^ monocytes during/after conversion. The presence of IMQ, in contrast, seems to allow for the generation of more Notch2 deficient cells, which is in line with our in vitro data showing that R848 treatment in the absence of Notch stimulation also increases Ly6C^lo^ cell numbers.

We also addressed the issue of cell death raised above. There was no evidence of increased cell death in mutant cells compared to wt cells in all relevant cell types analyzed (Figure 5L, M. See point 3 below). We have extended the Results section and Discussion section accordingly.

3) Lineage tracing is always challenging, and it is difficult to assess whether notch signaling is requested in the differentiation toward a specific lineage or is requested for cell survival. Notch regulated monocyte survival is a well taken point, since also Bcl2 is strongly decreased in Notch2-/- monocytes. Therefore, providing absolute numbers for the in vivo and in vitro experiments (cell numbers instead of %) is absolutely required. In fact, most of the study is based on % (often unclear among which population) which could lead to misinterpretation. The authors should provide absolute numbers per organs along with per mg, whenever possible. For example, "By comparison, the TLR4 ligand LPS also increased Ly6C^lo^ cell numbers and expression levels of Nr4a1and Pou2f2. However, the absolute conversion rate was lower under LPS and there was no synergy with DLL1 (Figure 1D and E)". The numbers are not evaluated here neither the conversion rate as long as survival difference cannot be excluded.

We now provide absolute cell numbers for all datasets in addition to rel. quantification (Figure 1B, D, F, G for in vitro studies. Figure 5H, Figure 3—figure supplement 3A-C for in vivo studies). In fact, for the in vivo data we had presented the data upon initial submission in Figure S5 and Figure 5K. In summary, the absolute cell numbers match the relative cell frequency analysis, thus supporting our conclusions.

With respect to survival, we present new data on cell death analysis (see above), which did not show evidence for increased cell death, supporting regulation of conversion as the main mechanism.

4) The 'unrestrained inflammation' part either should be experimentally proven our should be completely rephrased (or deleted). The authors state in the abstract that "the absence of functional Notch2 signaling promotes resident tissue macrophage gene expression (…) resulting in unrestrained systemic inflammation" that could be interpreted as an overstatement. The inflammatory response shows perturbation in Notch-deficient mice, but not a clear pro-inflammatory shift (see also point below). Accordingly, it is not clear, if the splenomegaly or the "unrestrained systemic inflammation" is directly caused by monocytes. LysM-Cre is also active in neutrophils, which similarly express high levels of Notch2 according to immgen and can contribute to the observed phenotype. If the authors want to keep the link of 'Notch2-deficient monocytes cause unrestrained systemic inflammation' then the authors should perform monocyte (anti-CCR2 treatment) and neutrophil (anti-Ly6G treatment) depletion experiments in IMQ-treated wt and Notch-deficient mice. If the observed splenomegaly in Notch2-deficient mice is reduced to wt levels when treated with CCR2 (but not after Ly6G treatment), it is likely that monocytes are the direct cause.

We have followed the advice and have altered the wording to ‘aberrant inflammation’, indicating that the mutant response differs from the wt response. See Results section and Discussion section. We also agree that the relevance of the changes is not clear in this somewhat artificial model. However, the phenomenon itself should be reported and the relevance and possible disease context should be studied in further studies. We have added this point to the Discussion section.

5) T0 purity and flow analysis (F4/80, Ly6C,CD43, CD11c CD11b and GFP) of the sorted monocyte from all mouse strains should be provided. Working with bone marrow monocytes can be precarious, as the bone marrow may be contaminated with progenitors (and should be mentioned in the text).

As requested, we have included the monocyte sorting strategy, T0 purity and analysis in Figure 1—figure supplement 1 A-D and Figure 2—figure supplement 1A and B. Overall, there is very high purity. Furthermore, our targeting strategy does not influence progenitors (see above and Gamrekelashvili, 2016), therefore, we suggest it is an unlikely confounder for the different outcomes between wt and ko after transfer. However, it could be a confounder concerning cell fates in general, e. g. DC-like populations in the spleen. We discuss the potential confounder of progenitor contamination in the text (Discussion section).

6) In Figure 3 the authors perform t-SNE analysis based on the Ly6C, CD43, MHCII, F4/80 and CD11c marker set and according to this identified 5 monocyte 'subsets' (Figure 2C). Please also show the corresponding flow cytometry analysis (dot plots) (especially for PB) to identify these 5 subsets by regular gating and to see intensity of especially F4/80 and MHCII staining in Ly6Chigh and Ly6Clow monocytes in all conditions. In addition, in the gating strategy Figure 3—figure supplement 21C the authors used F4/80 to discriminate CD115+ MF from monocytes. Rose et al., 2011 in Cytometry A showed that splenic monocytes also express F4/80 and that this antigen can be used to identify monocytes. Therefore, it is possible that MF cells in the authors' gating strategy are contaminated by (probably aged) Ly6C low monocytes that up-regulated F4/80. To counter argue this please show their gated MF in a FACS plot with Ly6C vs CD43.

Our initial t-SNE analysis was performed as a global screening for perturbations in the ‘monocytes pool’ (Live CD45^+^Lin^neg^CD11b^+^CX3CR1-GFP^+^) and we use this rather crude method to inform strategies for further analysis. However, the detailed subset analysis and quantification depicted in submission Figure 3E,F, Figure 5J,K and S3A-D had already been performed with a regular gating strategy shown in S2C.

Following the advice we now provide additional sets of analysis using regular gating in all conditions and show intensities of F4/80, MHC-II and CD43 (Figure 3—figure supplement 2A and B). We also provide FACS plots of cell populations and show Ly6C vs CD43 expression, which demonstrate that the expanded F4/80^hi^ population in mutant mice does not express the Ly6C^lo^ monocyte marker CD43 (Figure 3—figure supplement 2B). In addition, we demonstrate by flow cytometry expression of Mertk and CD64, prototypical macrophage genes, in Notch2 knock-out Ly6C^lo^ cells (Figure 5B-D). Furthermore, N2^ΔMy^ cells show enrichment of prototypical macrophage gene sets by GSEA (Figure 5A and Supplementary file 4 and Supplementary file 5). The combined data thus suggest at least partial acquisition of a macrophage phenotype by N2^ΔMy^ cells, and our argument is based on several lines of evidence. However, since the transition of monocyte to monocyte-derived macrophage state should be gradual process, and since we observed these cells mostly in the blood, which is an unusual place for classical MF but more in line with circulating monocytes, we changed the description to F4/80^high^CD115^+^ monocytes and to F4/80^high^MerTK^+^ (FM^+^) monocytes later in the text as suggested by the reviewer.

7) Consistent with the working hypothesis: Notch deficient LY6C^lo^ monocyte phenotype drastically changes following IMQ Figure 3. Why were Ly6C^lo^ wt and N2 deficient monocytes not examined without IMQ Figure 4? It is important to know how these cells alter from baseline and help distinguish if the effects are IMQ alone, Notch alone or IMQ and Notch.

This indeed is a limitation of our study (see reply to point no. 1).

8) The conclusion from these studies is in the presence of IMQ and absence of Notch2 LY6C^hi^ cells become more of a "macrophage" compared to the natural progression towards LY6C^lo^ monocytes Figure 5A/B. In Figure 5B, c and F what is the phenotype without IMQ. At present, without the controls, it is hard to comment on these experiments.

In our previous study (Gamrekelashvili et al., 2016) we have already addressed the differentiation of Ly6C^hi^ monocytes into Ly6C^lo^ monocytes under steady state conditions and have shown that this process is impaired in the absence of Notch2. In the steady state, Notch2 deficient mice mostly develop a MHCII+CD11c-CD43- population of monocytes, which do not express F4/80 (Figure 3—figure supplement 2B), MerTK or CD64 to a relevant extent (Figure 5B-D), and thus do not show the FM^+^ phenotype. These findings were recapitulated in adoptive transfer experiments at steady state (Gamrekelashvili et al., 2016) and with IMQ treatment (Figure 5E, G). Therefore, significant expansion of FM^+^ monocytes occurs only with IMQ treatment.

[Editors' note: further revisions were suggested prior to acceptance, as described below.]

Summary:All reviewers and editor agree that the authors have addressed most of the concerns that were raised and included important new data to support their hypothesis. Even though the authors could not perform the important bulk RNA-seq experiment of Notch-deficient Ly6C low monocytes under steady state conditions, they represented (and included in the manuscript) strong reasons for this (especially the mentioned selection bias). However, one last point on survival needs to be addressed and discussed in the revised version of the manuscript.Annexin V detection ex vivo cannot rule out the reduced survival of Ly6CClow monocyteApoptotic cells are not harvested for flow cytometry as the process is not synchronous. Hence, the hypothesis that Notch 2 control Mo survival should not be excluded. This alternative hypothesis is even supported by the absolute count where the number of missing Ly6Clow is not compensated by both MF and DC numbers in Notch 2 deficient in Figure 3. For instance, the difference of 200 cells/µl of Ly6chigh in the blood between WT and N2KO shows a difference of 50 cells /µl of Ly6C^lo^ at day 7 and a difference of 1.5 F4/80^hi^ cells /µl which rather argue for a defect in cell survival and a very weak accumulation of F4/80^hi^ cells in the blood. Similar discrepancy is observed in the spleen. Around 800,000 Ly6C^lo^ cell difference in the spleen for only an extra of 10,000 F4/80^hi^ cells.Absolute value for adoptive transfer experiment have not been provided. In addition, since bcl2 is strongly down in ko cells, a reduced survival does not necessarily mean a higher apoptosis rate.The authors need to discuss and state this survival point across the manuscript, whenever necessary.

In response to one reviewer, we added an extended discussion on the issue of cell survival. Over one paragraph, we now discuss the possibility of regulation of cell survival as an additional mechanism to explain the Notch2 loss-of-function phenotype:

In the case of Nr4a1 loss-of-function the reduced numbers of Ly6C^lo^ monocytes are due to increased apoptosis (Hanna et al., 2011). Although we did not find evidence for increased apoptosis due to Notch2-deficiency, our data do not exclude the possibility that regulation of cell survival by Notch2 contributes to the observed phenotype. In fact, two lines of the evidence suggest that regulation of cell survival might act synergistically to cell fate choices: first, in absolute numbers, there is no compensation by alternative cell fates for the number of lacking Ly6C^lo^ monocytes in the blood or spleen of N2^ΔMy^ mice; second, expression of Bcl2, a strong regulator of cell survival, is downregulated in IMQ-treated N2^ΔMy^ Ly6C^lo^ cells as compared to controls.